# Hyperthermia Treatment as a Promising Anti-Cancer Strategy: Therapeutic Targets, Perspective Mechanisms and Synergistic Combinations in Experimental Approaches

**DOI:** 10.3390/antiox11040625

**Published:** 2022-03-24

**Authors:** Ga Yeong Yi, Min Ju Kim, Hyo In Kim, Jinbong Park, Seung Ho Baek

**Affiliations:** 1College of Korean Medicine, Dongguk University, 32 Dongguk-ro, Ilsandong-gu, Goyang-si 10326, Korea; grapefruit9782@gmail.com (G.Y.Y.); 1996kimminju@naver.com (M.J.K.); 2Department of Surgery, Beth Israel Deaconess Medical Center/Harvard Medical School, Boston, MA 02215, USA; hyoin0428@gmail.com; 3College of Korean Medicine, Kyung Hee University, 24 Kyungheedae-ro, Dongdaemun-gu, Seoul 02447, Korea

**Keywords:** hyperthermia, cancer, combination therapy, chemotherapy, natural product, synergistic effect, reactive oxygen species, heat shock protein

## Abstract

Despite recent developments in diagnosis and treatment options, cancer remains one of the most critical threats to health. Several anti-cancer therapies have been identified, but further research is needed to provide more treatment options that are safe and effective for cancer. Hyperthermia (HT) is a promising treatment strategy for cancer because of its safety and cost-effectiveness. This review summarizes studies on the anti-cancer effects of HT and the detailed mechanisms. In addition, combination therapies with anti-cancer drugs or natural products that can effectively overcome the limitations of HT are reviewed because HT may trigger protective events, such as an increase of heat shock proteins (HSPs). In the 115 reports included, the mechanisms related to apoptosis, cell cycle, reactive oxygen species, mitochondrial membrane potential, DNA damage, transcription factors and HSPs were considered important. This review shows that HT is an effective inducer of apoptosis. Moreover, the limitations of HT may be overcome using combined therapy with anti-cancer drugs or natural products. Therefore, appropriate combinations of such agents with HT will exert maximal effects to treat cancer.

## 1. Introduction

Cancer remains the second biggest cause of death in North America after ischemic heart disease [1]. The incidence of cancer continues to rise despite the notable advances in screening, prevention, and treatment. Furthermore, the prevalence of cancer is expected to increase gradually because of the increased lifespan [2]. In 2020, Joannie Lortet-Tieulent et al. reported that high-income countries have low mortality rates for cancer because of early detection and appropriate therapies. On the other hand, low-income countries have higher mortality because of their late diagnosis. Furthermore, access to treatment is limited by the high expenses [3].

Although various treatments exist for cancer treatment and have been studied for a long time, conventional cancer treatment modalities have several flaws, such as recurrence, side effects, and high cost. For example, cachexia, caused by the side effects of certain chemotherapies, is associated with death. Thus, lesser side effects with low cost are attracting interest. Hyperthermia (HT) has begun to emerge as a promising strategy for cancer treatment to compensate for the shortcomings of conventional cancer therapies [4]. HT is a promising option because of its few side effects compared to conventional therapies and is cost-effective considering its large therapeutic gain achieved, especially when combined with other therapies. According to the Dutch Deep Hyperthermia Trial, HT showed enhanced effects when used with radiotherapy and the cost-per-life-year gained was less than 4000 Euros [5]. Moreover, consistent effort is being made to develop less expensive HT systems [6,7].

Clinical trials conducted to examine the effects of HT reported a significant reduction in the tumor size when HT is used in conjunction with other treatments. HT of 41–44 °C was not shown to be toxic to normal cells while inducing toxicity in cancerous cells [8]. Although not yet widely available, clinical application of HT is performed in 4 types: electromagnetic, focused ultrasound, hyperthermic perfusion, and conductive heating [9]. The methods are categorized based on the induction site: local, regional, and whole-body HT. Local HT can further be subcategorized into external, intraluminal/endocavitary, and interstitial HT; regional HT into deep tissue, regional perfusion, and continuous hyperthermic peritoneal perfusion. Induction sites are decided based on the type and region of cancer [10].

On the other hand, HT alone may not be effective enough to significantly destroy tumors. HT stimulation activates transcription factors involved with tumor survival, making tumor cells grow faster and resistant to cancer treatments [11,12]. To complement such characteristics, HT is generally used in conjunction with other therapeutic treatments such as radiation, chemotherapy, radiochemotherapy, gene therapy, surgery, and immunotherapy [10]. Recent studies have discussed the mechanisms of actions of the combination of HT with other cancer therapies. In particular, natural products can greatly benefit HT through induction of synergism with few side effects and low costs [4].

The main mechanisms of combined treatments include reactive oxygen species (ROS) production, DNA damage, cell cycle arrest, and mitochondrial membrane potential (MMP) depolarization, etc. These factors are related directly or indirectly to increased ROS levels. Increased ROS production in the mitochondria under conditions of heat shock results in nonspecific alteration of proteins, lipids, and nucleic acids, culminating in bioenergetic failure.

Here, we review experimental studies on HT in cancer. In vitro application was made by exposing cells to heat in a heated water bath or incubator system, and in vivo HT was performed either by water bath or clinical methods after modification. This review discusses the molecular mechanism of HT in cancer cells, which will help better understand the synergic effect on cancer cells when combined with other agents. Furthermore, effective HT combinations, including natural products and anti-cancer agents, are also analyzed to present a perspective to utilize HT as a novel anti-cancer therapy.

## 2. The Effect of HT Alone in Promoting Cell Death on Malignant Cells and the Mechanism

HT can induce apoptosis or other physiological changes in cancer cells. Jiang et al. investigated protein expressions in Tca8113 cells following HT at 43 °C for 40 min, utilizing fluorescent differential display by 2D gel electrophoresis. Changes were observed in 107 proteins, of which 57 were dramatically regulated 24 h after the HT treatment. The altered proteins were subcategorized into the following functional classes: energy metabolism-related enzymes, cytoskeleton-related proteins, chaperones, transcription factors and protein synthesis-related proteins, and cell division/proliferation-related proteins [13]. Cancer cells were not readily destroyed when HT was applied alone because of the above alterations. In some examples, elevated temperatures were used during a single session. In 2014, Takagi et al. reported that high-temperature HT (HTH) could be an established treatment option for cancer. This suggests that HTH at 60 and 70 °C suppresses tumor growth in a glioma rat model. In particular, cell proliferation was suppressed by HTH at 70 °C [14]. On the other hand, given that HTH is difficult to utilize in practice, this study focused on research with mild HT. HT often induces physiological changes that can also protect cancer cells from death. In a study by Shi et al., heat-treated melanoma cells at 42 °C for 4 h expressed increased levels of 70-kDa heat shock protein (HSP70). Enhanced cross-priming could be reproduced by the overexpression of HSP70 in melanoma cells transduced with HSP70 encoding lentiviral vector. Furthermore, HT increased the transcription of several tumor Ag-associated Ags, including AGE-B3, -B4, -A8, and -A10 [15]. Table 1 summarizes the studies of the effect and mechanisms of HT alone.

### 2.1. Reactive Oxygen Species (ROS) Production

HT is closely related to ROS production. Many studies have shown that the ROS level increases in tumor cells when HT is combined with other therapies. Few cases of increased ROS with HT alone have been reported. Hou et al. reported that HT at 43 °C for 60 min was associated with increases in the levels of intracellular ROS and caspase-3 activation in U-2 OS cells. Mitochondrial dysfunction was followed by the release of cytochrome c from the mitochondria. In addition, it was accompanied by decreased anti-apoptotic proteins, Bcl-2 and Bcl-xL, and increased pro-apoptotic proteins, Bak and Bax. The changes in these factors led to the apoptosis of U-2 OS cells and HT-triggered endoplasmic reticulum (ER) stress [16].

### 2.2. Heat Shock Proteins (HSPs)

HT can regulate HSPs when used alone. Kus-Liskiewicz et al. reported that hepatocytes could survive heat shock at 43 °C for 60 min, and heat shock factor 1 (HSF1) activation was surprisingly related to up-regulated gene expression. On the other hand, the induction of cell death by HSF1 is mediated by the simultaneous repression of several genes required for spermatogenesis. A broad range of other transcription factors activated by HSF1 could also contribute to cell-type-specific mechanisms regulating the transcriptional response to HT [17].

Previous studies showed that HT increases HSP expression. A remarkable study on the effect of photodynamic treatment, which works similarly to HT on HSP expression, was recently published. Frank et al. observed oxidative stress-related changes in the tumor tissues from a tumor model of Sprague-Dawley rats injected with DS-sarcoma ascites cells that was induced by localized HT at 43 °C for 60 min that was comparable to 5-aminolevulinic acid-based photodynamic therapy (ALA-PDT). Moreover, the combination with HT greatly improved the cancer apoptotic effects of ALA-PDT. The increase in HSP70 by the HT treatment was reduced by the ALA-PDT co-treatment, showing that HSP70 cannot protect tumors using this combination [18].

### 2.3. DNA Damage

Some mechanisms of HT are related to DNA damage. Oei et al. examined the effects of HT on HPV-positive cells using cervical cancer cell lines infected with HPV 16 (SiHa cells) and 18 (HeLA cells) using in vivo tumor models and ex vivo–treated patient biopsies. A clinically relevant HT treatment at 42 °C for 1 h resulted in early protein 6 (E6) degradation. E6 binds to p53 and mediates its ubiquitination and degradation [19]. HT-induced E6 degradation thereby prevented the formation of the E6–p53 complex and enabling p53-dependent apoptosis and G2-phase arrest [19]. In 2009, Deezagi et al. reported that HT induces apoptosis significantly in human myeloid leukemia cell lines, TF-1, K562, and HL-60. The cells exhibited a pre-apoptotic pattern at 41 and 42 °C after 60–120 min and an apoptotic pattern at HT at 43 and 44 °C for 30 min. The telomerase activity was stable immediately after HT at 41–42 °C for 60 min but decreased to 35–40% after 120 min [20]. Speit and Schutz reported that short-duration heat shocks and elevated temperatures over more extended periods induce chromosomal damage and inhibit DNA repair, leading to apoptosis of non-small cell lung carcinoma (NSCLC) A549 cells. HT-induced DNA damage was not rapidly removed by post-incubation at 37 °C but even increased after exposure to 48 °C for 60 min [21]. Barni et al. examined samples of myeloma cells immediately at the end of two HT treatments with a 15-day interval performed on a multiple myeloma patient. They reported significant degeneration, mostly demonstrating the features of apoptosis (cell shrinkage, DNA fragmentation, and karyorrhexis). The results indicated the maximal response at the end of the first hyperthermic treatment, with an approximately 19% increase compared to the control [22].

### 2.4. Cell Cycle Arrest

HT also can induce cell cycle arrest, which plays an important role in cancer apoptosis. Heat shock at 45 °C for 30 min initiated a process of rapid degradation in melanoma cells. Twenty-four hours of incubation decreased the cell viability, altered the cell morphology and F-actin cytoskeleton organization, and significantly reduced the number of adherent cells in B16-F10 melanoma cells. Most of the cells were in the late apoptosis state and showed an altered gene expression profile. A follow-up of two weeks after heat exposure showed that viability and number of adherent cells remained very low, with a high percentage of early apoptotic cells. Still, heat-treated cultures maintained a low but relatively constant population of cells in S and G2/M phases for an extended period after heat exposure [23].

### 2.5. Other Cellular Physiological Changes

#### 2.5.1. Cytoskeletal Alterations

One of the modes of mild HT-induced cell death in H1299 cells is the mitotic catastrophe. Twenty-four hours after a heat shock treatment at 43.5 °C and 45 °C for 30 min, the number of actin stress fibers was reduced significantly, microtubules formed a looser meshwork, a portion of the cells possessed multipolar mitotic spindles, whereas vimentin filaments collapsed into perinuclear complexes. These results mean that HT can induce cytoskeletal alterations [24]. Luchetti et al. reported using H1299 NSCLC cells that HT induces several cellular responses leading to morphological changes, cell detachment, and subsequent death. The loss of integrins from the cell surface by the acute heat treatment blocks several physiological signaling pathways. After HT at 43 °C for 1 h, the cytoskeletal proteins showed an increase of high-molecular-weight aggregates and a significant decrease in both actin and CD11a content with respect to control cells [25].

#### 2.5.2. Change in Expression of Genes

Changes in gene expressions on cell physiology after HT were also examined. Borkamo et al. detected reduced mRNA levels in a wide range of T-cell-, natural killer cell- and antigen-presenting cell-related genes after HT at 43 °C for 1 h. These data, obtained from BT4An-injected rats, showed that genes involved in protein dephosphorylation were overrepresented after HT, which results in cancer apoptosis. Among these genes was a group DUSPs (dual-specificity phosphatases) involved in the dephosphorylation of central vital proteins in signal transduction, such as the mitogen-activated protein kinase (MAPK) pathway [26]. Furusawa et al. observed apoptotic cells after 6 h of HT treatment at 42 °C for 90 min to U937 lymphoma cells. The elevated expression of HSPs, including HSP27, HSP40, and HSP70, was detected after the activation of HSF-1. In the cells treated with HT, 1334 probe sets were upregulated, while 4214 probe sets were downregulated by >2.0-fold. The up-regulated genes were related to cellular compromise (HSPA1B, DNAJB1, HSPH1, and TXN) and cell death (PML, LYN, and DUSP1). The down-regulated genes included CCNE1 and CEBPE, which are associated with cell proliferation [27].

#### 2.5.3. Damage of Collagen Fiber

HT can induce cancer cell apoptosis and necrosis through collagen fiber damage. In 2020, Piehler et al. reported that mild HT (40 °C and 42 °C) damages the collagen architecture in murine Achilles’ tendons. In addition, mild extrinsic (hot air) and iron oxide nanoparticle-based magnetic HT reduced the collagen fiber architecture in the generated hetero-type tumor spheroids [28].

#### 2.5.4. Cell Differentiation

Sharif-Khatibi et al. showed that heat treatment at 43 °C suppressed the growth of K562 cells without affecting their viability. HT at 43 °C induced hemoglobin synthesis and increased glycophorin A expression, indicating altered differentiation in K562 cells. HT of 43 °C induced HSP70 overexpression and protected the leukemia cells from apoptosis [29].

#### 2.5.5. Microvessel Damage

HT also causes psychological changes in cancer cells, such as microvessel damage. Li et al. investigated apoptosis and necrosis in murine H22 hepatoma cells after different thermal doses by dead cell analysis. Most of the dead cells were apoptotic in the initiation phase at the first time; the necrosis rates increased gradually. Microvessel damage following HT continued after the heat treatments. Most of the tumor cells were killed after microvessel damage in vivo. Therefore, the authors concluded that microvessel damage-induced cell death was an important mechanism of HT [30].

### 2.6. Regulation of Apoptosis Associated Transcription Factors and Proteins

#### 2.6.1. Transcription Factors

Kokura et al. reported that pretreating MKN45 cells with HT at 42 °C for 1 h significantly inhibited the tumor necrosis factor-alpha (TNF-α)-induced increase in the binding activity of nuclear factor-kappa B (NF-κB) to DNA, which results in cancer apoptosis [31]. In 2017, Chen et al. reported that HT with 42–46 °C for 1 h could inhibit proliferation, promote apoptosis, reduce the tumor formation rate, growth rate, angiogenesis rate, degree of hardness of tumors, ischemic tolerance and anoxic tolerance, and have synergy with temozolomide in C6 cells. HT produced these effects primarily by inhibiting the EGFR/STAT3/HIF-1A/VEGF-A pathway [32].

Wang et al. found that HT at 42 °C stimulated the release of TNF-α, decreased C6 glioma cell migration and invasive capability after 30, 60, 120, and 180 min of treatment, with increased spontaneous apoptosis in C6 glioma cells at 120 min. HT alone reduced the invasion of C6 glioma cells by stimulating TNF-α signaling to activate apoptosis, enhancing p38 MAPK expression, and inhibiting the NF-κB pathway [33].

Basile et al. reported that HT inhibits cell proliferation and induces apoptosis. In thermosensitive HepG2 cells, the induction of apoptosis and the involvement of apoptosis family genes Bax and Bcl-2 occur independently of p53. Therefore, in the thermoresistant HUT cells, the level of apoptosis is low, and the main effect of HT is the suppression of cell proliferation associated with a fast and durable overexpression of p53. Signaling by S100A4, given its ability to interact with p53 and the Notch pathways, also appears to be differentially operational during the induction of apoptosis. Apoptosis appears to be activated through alternative pathways independent of p53 [34].

Fukami et al. investigated the relationship between apoptosis-inducing factor (AIF) and apoptosis under various thermal conditions from 43 °C to 47 °C for 1 h using four p53-wild or -mutant human glioma cell lines. AIF translocation from the mitochondria to the nucleus under hyperthermal conditions was shown by confocal laser microscopy. The results showed that hyperthermal conditions induce AIF translocation and apoptotic cell death in the p53-mutant human glioma cells [35].

Wan and Wu explored the effects of HT on HIF-1α expression, proliferation, and lung cancer angiogenesis using NSCLC and small cell lung cancer (SCLC) cell lines. After HT at 47 °C for 40 min, the proliferation and angiogenesis potentials of residual NSCLCs and SCLCs are induced by HIF-1a. On the other hand, HIF-1α expression in NSCLCs is regulated by both the AKT and ERK signaling pathways, but HIF-1a expression in SCLCs is regulated only by the AKT signaling pathway. As a result, HT leads to cancer apoptosis through the AKT and ERK signaling pathways [12].

In 2005, Pajonk et al. examined the effects of HT at 44 °C for 1 h on proteasome function. Its significance for signal transduction, cell death, and androgen receptor (AR) expression was studied in PC-3, LnCaP, and DU-145 human and TRAMP-C2 murine prostate cancer cells. HT caused apoptosis and radiosensitization and decreased the 26S proteasome activity in all three human cell lines to ~40% of untreated control cells. The heat treatment inhibited the constitutive and radiation-induced activation of NF-κB caused by the stabilization of IκB. The AR protein levels in LnCaP cells, an important pathway of androgen-dependent prostate cancer, decreased dramatically after heat treatment [36].

#### 2.6.2. Regulation of Anti/Pro-Apoptotic Protein

Morle et al. suggested that c-FLIP is a thermosensitive protein that is a target of HT. HT-mediated regulation of c-FLIP allows restoration of apoptosis in human cancer cell lines (MDA-MB-231, A549, SK-HEP-1, DU145, HCT116 and SW480 cells) induced by TNF ligands, including TNF-related apoptosis-inducing ligand (TRAIL). HT at 42 °C for 1 h induced c-FLIP depletion from the cytosolic fraction, without apparent degradation, thereby preventing c-FLIP recruitment to the TRAIL DISC and allowing efficient caspase-8 cleavage and apoptosis [37]. In 2011, Xie et al. reported that HT at 43, 45, and 47 °C for 30 min could effectively inhibit tumor invasion in vitro with temperature dependence. The down-regulation of transforming growth factor-beta 1 (TGF-β1), vascular endothelial growth factor (VEGF), and MMP-2/9 of MCF-7 cells by HT treatment played an important role in the effect of suppressing invasion ability and apoptosis of residual cancer cells [38]. Zhou et al. reported that heat treatment at 45 °C for 40 min induced the apoptosis and necrosis of CaSki cells. During apoptosis, the caspase-3 and Smac levels were up-regulated, and anti-apoptotic Survivin was down-regulated, resulting in enhanced programmed cell death [39]. In Tca8113 cells, significant cell apoptosis was observed after HT at 43 °C for 80 min. Marked PKC-δ cleavage fragmentation was also identified. Among the proteins interacting with PKC-δ, 39 and 16 were involved in promoting and inhibiting the apoptosis of Tca8113 cells, respectively [40]. Yasumoto et al. found that the SAS/neo (wild-type p53; wtp53) cells were sensitive to heat at 44 °C for 40 min. The activation of caspase-3 and apoptosis in the wtp53 cells was clearly higher than those in SAS/mp53 (mutated p53; mp53) cells. The expression of apoptosis-suppressive genes, such as IL-12p35, decreased in the wtp53 cells while IL-12 Rβ1 increased in mp53 cells, even though apoptosis-promotive genes, such as caspase-9, CD30, and CD40, were increased independently of p53 by HT [41].

### 2.7. Activation of Caspase-3

Shellman et al. reported that a HT treatment at 45 °C and 48 °C for 90 min induces apoptosis in a non-conventional manner that is not through the extrinsic or intrinsic pathways but rather through ER-mediated apoptosis. This discovery was reported in skin cancer cell lines, both melanoma and non-melanoma. HT did not activate caspases-8 or 9 but activated caspase-3/7, suggesting a non-conventional apoptotic pathway. An analysis of Grp78 expression and caspase-12 or 4 activation indicated that HT could induce ER-mediated apoptosis [42].

Vertrees et al. reported a 5.7-fold increase in TRAIL and insignificant changes in TNF-α, FAS-L, and caspases-3, 8, 9 in transformed cells (isogenic human lung cancer cell line transformed with H-ras) after HT at 43 °C for 3 h. This study reported the upregulation of TRAIL in H-ras transfected cells, cancer apoptosis, and a disruption in the MAPK pathway that partially fitted this requirement [43]. In 2015, Qin et al. found that HT at 43 °C for 40 min decreased the expression of Survivin, prevented its binding to X-linked inhibitor of apoptosis protein (XIAP), activated caspase-3, and induced apoptosis in EC109 esophageal cancer cells [44]. Mantso et al. showed that low HT (43 °C) triggered extrinsic and intrinsic apoptotic pathways, both apoptosis pathways associated with the activation of caspase-6 in human malignant melanoma cells A375 and A431. After high HT (45 °C), caspase-3, 7, and 6 worked together. Furthermore, there were significant activities in the ER under both HT, suggesting that ER plays an essential role in HT-induced apoptosis through the activation of CCAAT/enhancer binding protein homologous protein (CHOP). Inositol-requiring enzyme 1 and activating transcription factor 6 (ATF 6) were activated by low HT, but PERK was also activated by high HT, suggesting that ER stress can induce CHOP through different transmitting signals [45].

**Table 1 antioxidants-11-00625-t001:** **1-1.** Effect of sole HT on tumor cells (unknown mechanism). **1-2.** HT induces ROS production in tumor cells. **1-3.** HT regulates HSPs in tumor cells. **1-4.** HT induces DNA damage in tumor cells. **1-5.** HT induces cell cycle arrest in tumor cells. **1-6.** HT induces cytoskeletal alterations in tumor cells. **1-7.** HT induces change in expressions of tumor cell genes. **1-8.** HT induces damage of collagen fibers in tumor cells. **1-9.** HT induces cell differentiation in tumor cells. **1-10.** HT induces microvessel damage in tumor cells. **1-11.** HT regulates apoptosis-associated homeostasis in tumor cells.

**1-1.**			
**HT**	**Cell Line and Observation Model**	**Molecular Mechanism**	**Ref**
43 °C, 40 min	Human tongue squamous cell carcinoma, Tca8113/in vitro	Apoptosis	[13]
50–70 °C	Glioma/in vivo	Necrosis, Apoptosis	[14]
42 °C, 4 h	Melanoma, HLA-A*0201^+^ Me275, Me290/in vitro	via up-regulation of both HSPs and tumor Ag expression	[15]
**1-2.**			
**HT**	**Cell Line and Observation Model**	**Molecular Mechanism**	**Ref**
43 °C, 1 h	Human osteosarcoma, U-2/in vitro	Increasing ROS and caspase-3 activation, releasing of cytochrome c and ER stress	[16]
**1-3.**			
**HT**	**Cell Line & Observation Model**	**Molecular Mechanism**	**Ref**
43 °C, 1 h	Spermatocytes/in vivo	Deregulated RNA (especially piRNA) metabolism	[17]
43 °C, 1 h	Sarcomas/in vivo	Increased protein nitration/Decreased GSH levels and hsp 70 expression	[18]
**1-4.**			
**HT**	**Cell Line and Observation Model**	**Molecular Mechanism**	**Ref**
42 °C, 1 h	Cervical carcinoma, SiHa, HeLa, C33A, Caski, C4I, HT3/Human prostate carcinoma, Du145, LNCaP, PC3/in vitro	E6 degradation enabling p53-dependent apoptosis and G2-phase arrest	[19]
41–44 °C, 2 h	Human myeloid leukemia, TF-1, K562, HL-60/in vitro	Decreasing of telomerase activity	[20]
42–48 °C, 30 min–2 h	NSLCLs, A549/in vitro	Inducing DNA damage, chromosomal damage and to inhibit DNA repair	[21]
41 and 43 °C, 90 min	Myeloma/in vitro	Cell shrinkage, DNA fragmentation, karyorrhexis	[22]
**1-5.**			
**HT**	**Cell Line and Observation Model**	**Molecular Mechanism**	**Ref**
45 °C, 30 min	Melanoma, B16-F10/in vitro	G0/G1 arrest	[23]
**1-6.**			
**HT**	**Cell Line and Observation Model**	**Molecular Mechanism**	**Ref**
43.5 and 45 °C, 30 min	Lung cancer, H1299/in vitro	Cytoskeletal alterations	[24]
43 °C, 1 h	Human neuroblastoma, SK-N-MC/in vitro	Loss of integrins from the cell surface, that blocks several physiological signaling pathways	[25]
**1-7.**			
**HT**	**Cell Line and Observation Model**	**Molecular Mechanism**	**Ref**
43 °C, 1 h	Glioma BT4An tumor/in vivo	Changes of global gene expression	[26]
41 °C, 30 min	Human lymphoma, U937/in vitro	Change in the expression of a large number of genes such as DNAJB1, HSPA1A, and HSPA1B	[27]
**1-8.**			
**HT**	**Cell line and Observation Model**	**Molecular Mechanism**	**Ref**
40 and 42 °C, 1 h	Pancreatic cancer, Panc-1 and fibroblast, WI-38/in vitro	Affecting collagen fiber architecture and inducing apoptosis	[28]
**1-9.**			
**HT**	**Cell Line and Observation Model**	**Molecular Mechanism**	**Ref**
43 °C (30, 60, 90 min),45 °C (20, 40, 60 min)	Erythroleukemia, K562/in vitro	Erythroid differentiation (inducing glycophorin A expression, hemoglobin synthesis)	[29]
**1-10.**			
**HT**	**Cell Line and Observation Model**	**Molecular Mechanism**	**Ref**
43 °C, 30 min	Hepatoma, H22/in vivo	Apoptosis, Necrosis/Inadequate supply of nutrients, oxygen, and accumulation of acid	[30]
**1-11.**			
**HT**	**Cell Line and Observation Model**	**Molecular Mechanism**	**Ref**
42 °C, 1 h	Human gastric cancer, MKN45/in vitro	Inhibiting the TNF-alpha-induced NF-kappaB activation	[31]
43 °C, 1 h	Glioblastoma, C6/Human umbilical vein endothelial cells (HUVECs)/Human renal tubular epithelial cells, HK2/in vitro	Inhibiting proliferation and promoting apoptosis through the EGFR/STAT3 pathway	[32]
42 °C, 0–4 h	Glioma, C6/in vitro	Stimulating TNF-α signaling to activate apoptosis, enhancing p38 MAPK expression and inhibiting the NF-κB pathway	[33]
42 °C, 1–48 h	Hepatocellular carcinoma, HepG2, HUT/in vitro	Increased expression of Bax and down regulation of Bcl-2 and S100A4 genes	[34]
43, 45, and 47 °C, 1 h	Human glioma, A172, T98G, U251MG, YKG-1/in vitro	A temperature-dependent AIF translocation that can cause apoptosis independent of p53	[35]
47 °C, 40 min	Human NSCLC, NCI-H1650/SCLC, NCI-H446/in vitro	Regulation of HIF-1a expression through AKT and ERK signaling pathways.	[12]
44 °C, 1 h	Human prostate cancer, PC-3, LnCaP, DU-145/Murine prostate cancer, TRAMP-C2/in vitro	Inducing proteasome inhibition and loss of androgen receptor expression, abrogating AR expression, down-regulation of NF-kB	[36]
42 °C, 1 h	Breast cancer, MDA-MB-231/in vitro	Aggregation-induced c-FLIP cytosolic depletion	[37]
43, 45, and 47 °C, 30 min	Breast carcinoma cell, MCF-7/in vitro	Down-regulation of the expression of TGF-β1, EGF and MMPs, suppressing MMP-2/9 secretion and enzymatic activity	[38]
45 °C, 40 min	Human cervical cancer, CaSki/in vitro	Apoptosis, Necrosis/Up-regulation of caspase-3 and Smac levels and down-regulation of anti-apoptotic Survivin,	[39]
43 °C, 80 min	Human tongue squamous carcinoma, Tca8113/in vitro	Activation and translocation of PKC-δ	[40]
44 °C, 40 min	Human squamous cell carcinoma (SAS) wild-type 53 and mutated-type 53/in vitro	Induction of caspase-3 activation and apoptosis in the wild type p53 and suppression of IL-12-related genes in the mutated p53	[41]
45 °C, 48 °C, 90 min	Human melanoma, A375/Squamous carcinoma, A431/in vitro	Apoptosis (45 °C), Necrosis (48 °C/Activation of caspase-3/7, ER stress and ER-mediated apoptosis	[42]
43 °C, 3 h	Lung cancer, BEAS-2B and BZR-T33/in vitro	Increasing caspase-3 as a result of activation of cell-death membrane receptor, arrest of cells in the G2-Mphase of the cell cycle	[43]
43 °C, 40 min	Esophageal cancer, EC109/in vitro	Inhibiting Survivin and XIAP and activating caspase 3	[44]
43 °C, 45 °C, 2 h	Human malignant melanoma, A375, A431/in vitro	low HT: induced extrinsic and intrinsic apoptotic pathways both of which activated caspase 6 only//high HT: mediated by the combined effects of caspases 3, 7 and 6	[45]

## 3. Combination of Hyperthermia and Anti-Cancer Drugs

Anti-cancer drugs are the most evident and powerful treatments of cancer. There have been steady research attempts to utilize combined treatments of HT and cancer drugs to compensate for the limitations of chemotherapy, such as side effects, drug resistance, and recurrence. This section and Table 2 summarizes the studies on the combined treatment of HT and anti-cancer drugs.

### 3.1. Cisplatin

Cisplatin (cis-diamminedichloroplatinum [II]; CDDP; Platinol) is one of the best-known chemotherapy drugs. It is a white or deep yellow to yellow-orange crystalline powder at room temperature. Cisplatin has been proven to treat various types of cancers, including sarcomas, ovaries, bones, muscles, and the head. It was the first anti-cancer platinum compound approved by the United States in 1978 [46]. Studies have shown that cisplatin attacks more than one place simultaneously, causing the anti-cancer effects. The primary mechanism of action can be described as follows: binding with genomic DNA (gDNA) or mitochondrial DNA (mtDNA) to produce DNA lesions; blocking the production of DNA, mRNA, and proteins; arresting DNA replication; activating several transduction pathways that finally lead to necrosis or apoptosis [47]. Cisplatin is a first-generation anti-cancer drug with more significant side effects than recently developed drugs. Therefore, studies have focused on enhancing its effects at lower concentrations. In particular, the application of HT in parallel with cisplatin has been studied.

The combination of cisplatin and HT synergistically promotes apoptosis and inhibits cancer progression. Zhang et al. reported that the co-treatment reduced the cell viabilities and colony formation and induced apoptosis in PC-3 and DU-145 cells. The activation of caspase-3, cleavage of poly (ADP-ribose) polymerase (PARP), and increased AMP-activated protein kinase alpha (AMPKα) and c-Jun N-terminal kinase (JNK) signaling pathway were observed while suppressed Akt-mTOR-p70s6k signaling pathway was seen [48]. SESN1, an anti-apoptotic factor, is induced by a cisplatin treatment in IMC-3CR cells and is expected to be the main reason for cisplatin resistance in head and neck cancer. Narita et al. reported that the knockdown of SESN1 by RNAi reversed the resistance, and apoptosis was induced with a co-treatment of HT at 44 °C for 30 min and cisplatin. SESN1 also protects cells from ROS-mediated injury, another anti-cancer mechanism of cisplatin or HT treatment. Therefore, the repression of SESN1 enhanced the apoptotic effect via the ROS-caspase pathway of the co-treatment of cisplatin and HT in IMC-3 and IMC-3CR cells [49]. In 2014, mild HT (42–43 °C) was applied to gold nanorod (GNR). GNR-mediated mild HT increased cisplatin cytotoxicity, leading to the promotion of apoptosis in SKOV3 cells in vitro. In vivo studies with a xenograft model showed that the combined treatment delayed tumor development [50].

Cisplatin liposome formulations show less systemic toxicity than the free medication. In addition, administering cisplatin in liposomes results in selective delivery to solid tumor locations. The A549 and H460 xenograft models were studied using cisplatin liposome formulations. A heat-activated thermosensitive liposome formulation of cisplatin with a combination of HT was more effective than the free drug or a non-thermosensitive liposome formulation of cisplatin (i.e., Lipoplatin) [51]. Ohtsubo et al. suggested a temperature-time condition of HT by showing that the combined therapy of cisplatin with HT in 44 °C for 30 min was more effective than a co-treatment with 42 °C for 105 min. The co-treatment with 44 °C-30 min and cisplatin produced a higher apoptosis rate, DNA fragmentation, and poly (ADP-ribose) polymerase cleavage in IMC-3 cells [52].

Cisplatin and carboplatin are platinum analogs, one of the most widely used chemotherapy drugs in cancer treatment. Carboplatin is less effective than cisplatin in treating certain types of cancer [53], but apoptosis during the G1 phase is induced when HT (42 °C) is combined compared to HT or carboplatin single treatments, which G2/M arrest. The activation of caspase 9 and PARP cleavage is observed, and cytochrome C is also released in WERI cells [54].

### 3.2. Cisplatin and 5-Fluorouracil

5-Fluorouracil (5-FU) is another widely used cancer medicine. In particular, it has a significant anti-cancer effect in colon cancer by inhibiting thymidylate synthase and dysfunction of its metabolites into RNA and DNA [55]. Shi et al. designed a study to determine the effect of a combination with RF (radiofrequency)-HT (42 °C), cisplatin, and 5-FU on esophageal cancer. In vitro apoptosis analysis using human esophageal squamous cancer cells showed that the combinational therapy group had more apoptotic cells than the other treated groups. In vivo studies of nude rats injected with esophageal cancer cells also showed the highest suppression in cancer growth by the co-treatment [56].

### 3.3. Cisplatin and Doxorubicin

Doxorubicin, also known as adriamycin, is another anti-cancer drug used frequently in combination with cisplatin. Doxorubicin is a type of anthracycline anticancer drug that treats various types of cancer by inhibiting DNA topoisomerase II, intercalating DNA, and producing free radicals [57]. Schaaf et al. reported that a co-treatment of doxorubicin, cisplatin, and HT in 42 °C for 1 h induced DNA damage in ovarian and colon cancer. Cisplatin and doxorubicin-induced DNA damage. HT inhibited repair of DNA damage through PARylation and thus prevented replication. Therefore, together with HT, doxorubicin and cisplatin increased double-strand breaks and damage in BRCA-competent cells, OVCAR8, OVCAR3, HCT116 and LOVO cells [58].

### 3.4. Cisplatin and Ferucarbotran

Ferucarbotran consists mainly of a hydrophilic colloidal solution of superparamagnetic iron oxide coated with carboxydextran. It is commonly used as a magnetic resonance imaging contrast agent because of its low cost [59]. Sato et al. examined apoptosis of cancer cells by the co-treatment of cisplatin, ferucarbotran (Resovist^®^), and HT in oral cancer cells (HSC-3 and OSC-19). Cisplatin alone caused apoptosis by decreasing the G0/G1 phases and increasing cells in the S phase. Ferucarbotran generated an iron concentration sufficient to increase the temperature to 42.5 °C upon exposure to AMF (alternating magnetic field). A synergistic effect on apoptosis was observed when combined with ferucarbotran/AMF-induced HT and cisplatin [60].

### 3.5. Cisplatin + Sodium Arsenite

Arsenic has been used as a treatment in the past despite the significant hazard. Arsenic sensitizes cancer to HT, radiation, cisplatin, adriamycin, doxorubicin, and etoposide. As_2_O_3_ (trisenox) is an approved essential drug in leukemia management [61]. Muenyi et al. reported induced apoptosis of A2780 and CP70 cells by a combined treatment with sodium arsenite (NaAsO_2_), cisplatin, and HT. Arsenite combined with HT increased the response of p53-expressed ovarian cancer cells to cisplatin. Subsequently, cell death occurred by the disrupted cell cycle with G2 arrest and accumulation in pseudo-G1 [62].

### 3.6. Paclitaxel and Cisplatin

Huang et al. reported that Fas expression was increased in OS cells upon HT of 43 °C for 1 h with a combined treatment of paclitaxel and cisplatin compared to their individual applications, indicating HT-induced sensitization to chemotherapy. The survival rate of OS cells with HT of 43 °C for 1 h following a co-treatment with 10 g/mL paclitaxel and 5 g/mL cisplatin was 11.96%, which was significantly lower than when either 10 g/mL paclitaxel (45.02%) or 5 g/mL cisplatin (48.69%) was used alone [63].

### 3.7. Paclitaxel 

Paclitaxel belongs to a class of taxanes and is an anti-neoplastic drug that affects the stabilization of microtubules. Paclitaxel acts through various ways to induce cell death, such as specific cellular processes, the modulation of immune response via regulation of chemokines, cytokines, or immune cells [64]. 

In 2019, synergistic suppression was identified in a combination of microwave-induced HT at temperatures below 41 °C and paclitaxel on the growth of the human breast cancer cell line, MCF-7. The combined treatment of paclitaxel and 2 h of HT at 40.5 °C synergistically increased apoptosis. In addition, a combination of paclitaxel and HT with a temperature of 41 °C increased the apoptotic ratio from 12.21 ± 1.02% to 16.36 ± 2.39% [65]. Michalakis et al. showed that HT displayed cytostatic effects on all cell lines, especially at 43 °C in MCF-7 cells. The cytotoxicity was increased when HT was combined with paclitaxel. The 41.5 °C HT changed cell cycle distribution arrested at the G2/M phase, and co-treatment with paclitaxel affected the cell cycle kinetics in MCF-7 and SKOV-3 cells [66]. Lin et al. used short-term HT. This study subjected cells to thermotreatment maintained at 41 °C for 2 h, followed by a temperature drop to 37 °C. The cells were cultured routinely for 22 h. Short-term HT increased the sensitivity of MCF-7 cells to paclitaxel and reduced the IC_50_ of paclitaxel from 18.2 ± 1.0 to 15.0 ± 0.45 nm (*p* < 0.05). The apoptosis promoted by activated caspase-7 and arrested cell cycle in the G2/M phase were observed [67].

Furthermore, HT can inhibit mitosis and reduce the chemoresistance to paclitaxel. Giovinazzi et al. reported that the supplemental treatment of HT to paclitaxel-treated cells triggered mitotic slippage through the rapid decline of the cyclin B levels. Similarly, HT induced mitotic exit in cells blocked in mitosis by other anti-mitotic drugs. As a result, the apoptotic ratio of the combination of paclitaxel and HT was significantly higher in HepG2 cells [68]. In 2005, Michalakis et al. reported that when MCF-7 cells were treated with various amounts of paclitaxel, most notably at 10 μM, they were partially stopped at the G2/M phase. Paclitaxel combined with heat (41.5 °C for 2 h) resulted in mitotic catastrophe and widespread necrosis [69].

### 3.8. Docetaxel

Like paclitaxel, docetaxel belongs to a taxane. Docetaxel is a second-generation semi-synthetic taxane extracted from the European yew tree. The drug has an anticancer effect through binding and stabilizing tubulin. It disturbs microtubule disassembly, arrests the cell cycle at the G2/M phase, and induces cell death [70]. The combination of mild HT (41 °C, 2 h) with docetaxel was reported to increase the apoptosis rate compared to the docetaxel only group from 23.66 ± 3.59% to 47.12 ± 6.73% in MCF-7 cells and from 18.51 ± 3.17% to 55.16 ± 7.42% in MDA-MB-453 cells. The co-treatment arrested the cells at the G2/M cell cycle and decreased the Bcl-2/Bax expression ratio to induce apoptosis and suppress tumor growth [71].

### 3.9. Doxorubicin

Doxorubicin is an anthracycline antibiotic isolated from the Streptomyces peucetius species and has been used to treat various cancers. The compound intercalates into DNA, inhibits topoisomerase II, and produces free radicals [72].

In 2011, a mesoporous silica nanocontainer Si-SS-CD-PEG was used to carry doxorubicin to its specific target and induce cell death. HT increased the glutathione (GSH)-mediated release of doxorubicin from doxorubicin-loaded Si-SS-CD-PEG. HT at 42 °C increased the release of doxorubicin slightly, and notably increased the GSH content in A549 cells. The combination of HT at 42 °C with the carrier significantly increased clonogenic death and apoptosis [73]. Chae et al. examined the efficacy of systemic doxorubicin with pulsed high-intensity, focused ultrasound-induced, localized mild HT (HIFU-HT) to determine the combined effect of HT and doxorubicin in vitro and in vivo. In vitro, nuclear penetration of doxorubicin was stronger at 42 °C HT than at 37 °C, and was increased in a time-dependent manner. The treatment of HIFU-HT plus doxorubicin inhibited the growth of cancer cells the most compared to the other groups in vivo. HIFU-HT was induced by HIFU after optimizing the parameters with direct temperature measurements (frequency = 1 MHz, pulse repetition frequency = 5 Hz, power = 12 W, duty cycle = 50%) [74]. Vancsik et al. reported that, using C26 murine colorectal cancer cells, modulated electro HT (mEHT) induced apoptosis while doxorubicin mainly caused tumor cell necrosis. mEHT is a noninvasive locoregional supplement to radiotherapy and chemotherapy that has been applied effectively in various combinations to treat human gliomas and soft tissue sarcomas. Moreover, mEHT (42 °C, 2 h) promoted doxorubicin absorption to cancer cells, and the co-treatment with doxorubicin increased cell apoptosis. In addition, the combined treatment with HT and doxorubicin initiated caspase-dependent apoptosis and p21waf1 mediated growth arrest pathways [75]. Liang et al. investigated ‘smart’ gold nanoshells (GNs) made from gold nanoparticles, doxorubicin, and the targeting peptide A54. GNs with near-infrared (NIR) plasmon resonance generated localized heat-induced HT and served as a nanoblock. The combination treatment of doxorubicin/A54@GNs in 43 °C increased the doxorubicin concentration and induced cell apoptosis in BEL-7402 [76]. Jeon et al. loaded doxorubicin into ferucarbotran (Resovist^®^) via ionic interactions with the anionically charged carboxydextran coating layer of Resovist. The apoptosis rates of the tumors were 11.52 ± 3.10% in group A (normal saline, n = 4), 23.0 ± 7.68% in B (doxorubicin, n = 5), 25.4 ± 3.36% in C (MNP, n = 5), and 39.0 ± 13.2% in D (MNP/doxorubicin complex, n = 5) measured by bioluminescence imaging (BLIs), respectively. These in vivo results from Hep3B-injected mice showed that the Resovist/doxorubicin co-treatment induced higher apoptosis than sole therapies [77]. In 2013, Tang and McGoron combined doxorubicin with two HT methods, including rapid HT achieved by a NIR laser and slow rate HT provided by a cell culture incubator. The rapid HT caused cell necrosis, and the slow rate HT induced mild apoptosis. The co-treatment of doxorubicin with HT also overcame the P-gp mediated multidrug resistance of MES-SA in vitro [78]. 

Tang et al. used IR-780-loaded pH-responsive polymeric prodrug micelles with NIR with doxorubicin to overcome drug resistance by changing the permeability and fluidity of the plasma membrane. The polymeric prodrug micelles combined with the NIR laser irradiation increased intracellular doxorubicin accumulation significantly and synergistically induced cell apoptosis in doxorubicin-resistant MCF-7/ADR cells. Furthermore, tumor growth was inhibited in MCF-7/ADR tumor-bearing mice [79]. Liu et al. reported the highest level of cytotoxicity in 4T1 murine breast cancer cells treated with a water-responsive phospholipid-calcium-carbonate hybrid nanoparticle loaded with doxorubicin and indocyanine green (PL/ACC-DOX&ICG) and irradiated with a NIR laser in vitro. The combined treatment with PL/ACC-DOX&ICG and NIR laser irradiation produced the most suppression, resulting in a decrease in the multicellular tumor spheroid (MCTS) volume. Indocyanine green-induced HT generated thermal ablation, enhanced the doxorubicin uptake, and penetrated the tumor locations. Under NIR laser irradiation, PL/ACC-DOX&ICG had an effective photothermal effect and displayed considerably enhanced anti-cancer effects [80].

Pegylated liposomal doxorubicin (PLD) is a complex doxorubicin formulation based on pharmaceutical nanotechnology exhibiting novel pharmacokinetics and pharmacodynamic properties. PLD has a prolonged circulation time with consistent payload retention, as well as accumulation in tumors with high vascular permeability. As a result, the treatment effect conferred significant advantages over the conventional treatment [81]. In liposomal medications, ATP-binding cassette (ABC) transporters exhibited increased cellular absorption and decreased efflux. Wu et al. used a brain metastasis in vivo model of breast cancer, showed that a 10 min of focused ultrasound HT increased blood–brain tumor barrier permeability and blood perfusion in tumors significantly, thereby increasing nano-drug delivery into the interstitial tumor tissues and decreasing tumor growth [82].

Mild HT at 42 °C can exaggerate the effects of chemotherapy by rupturing the tumor tissue, enhancing cellular metabolism, cytoplasmic membrane permeability, and hastening tumor cell death. Gao et al. reported that intratumoral radiofrequency-induced HT at 42 °C enhanced the efficiency of liposomal doxorubicin in human HCC cancer HepG2 cells, which could be evaluated in vitro and in vivo using bioluminescence optical and ultrasonic imaging [83]. 

Eetezadi et al. reported that the block copolymer micelle (BCM)-doxorubicin formulation was capable of encapsulating doxorubicin at concentrations ranging from 2 to 7.6 mg/mL, depending on the copolymer to drug ratio used in a 2D and 3D cell culture of the human ovarian cancer cell lines HEYA8, OV-90, and SKOV3. Compared to PLD, BCM-doxorubicin led to stronger short-term suppression of MCTS growth, even comparable to long-term inhibition [84]. Current clinical research has shown that medications that use liposomes as carriers have benefits, such as immunogenicity, low toxicity/adverse reactions, and high adsorption efficiency [85]. Tsang et al. used liposome-encapsulated doxorubicin (Lipodox^®^) to enhance the drug’s killing effect on tumor cells by trapping doxorubicin within the tumor site. Macropinocytosis using mEHT at 42 °C for 30 min stimulated particle absorption. mEHT-enhanced Lipodox^®^ absorption increased the therapeutic impact and apoptosis of CT26 cells, both in vitro and in vivo [86].

Li et al. showed that the HT induced by gold nanorods (AuNRs) in NIR successfully erased the drug resistance in MCF-7 ADR cells, significantly increasing the effects of doxorubicin. The intracellular doxorubicin distribution of AuNR-loaded hyaluronic acid nanogels with doxorubicin (HA-CysNG@AuNR) was examined by measuring the fluorescence intensity. The fluorescence intensity increased with time, indicating that more doxorubicin-loaded nanogels were incorporated into the MCF-7 cells. At an intracellular GSH concentration of 10 mM, disulfide bond breaking enhanced doxorubicin release and cell death [87].

### 3.10. Bortezomib

Bortezomib is a first-in-class proteasome inhibitor approved to treat multiple myeloma and mantle cell lymphoma after prior therapy. Proteasome inhibitors disrupt various pathways in the cancer and cancer microenvironment, causing apoptosis and inhibiting cell cycle progression, proliferation, and angiogenesis [88]. The blocking of proteasome induced G0/G1 cell cycle arrest and increased regulatory proteins, contributing to onco-pathogenesis [89].

Mantle cell lymphoma (MCL) cell lines are sensitive to bortezomib and are thermosensitive with a low basal expression of HSP27. HSP27-overexpressed, bortezomib-resistant MCL cells show increased thermo-resistance. Milani et al. reported that a HSP27 blockade sensitized the resistant MCL cells to heat and chemotherapy, particularly bortezomib [90]. In 2007, Saliev et al. found that the combination of HT (44 °C, 15 min) and bortezomib improved apoptosis induction in U937 cells, increasing the percentage of cells reaching the late apoptotic state. Simultaneously, similar treatment for peripheral blood mononuclear cells (PMBCs) induced early apoptosis [91].

### 3.11. Gemcitabine (GEM)

GEM is a pyrimidine antimetabolite that is frequently used in systemic cancer treatment [92], specifically used to treat ovarian, breast, NSLCL, and pancreatic cancers among other solid tumors [93].

As shown in Adachi et al.’s research, HT inhibits gemcitabine-induced NF-κB activation. Furthermore, this study demonstrates that the combination of HT (43 °C, 1 h) and GEM resulted in enhanced cytotoxicity against AsPC-1 and MIAPaCa-2 cells. Additionally, HT, rather than gemcitabine, increased HSP70 levels [94]. Jin et al. showed that the majority of pancreatic cancer cells were arrested in the S-phase rather than entering the G2/M stage after being treated with GEM and heat at 42 °C for 90 min. HT increased the sensitivity of pancreatic cancer SW1990 cells to GEM by reducing cell growth and triggering cell death via ROS/JNK signaling [95]. Vertrees et al. reported that a HT co-treatment (43 °C, 3 h) in BZR-T33 cells increased gemcitabine-induced G2-M arrest, resulting in cell death. The combined therapy also induced a considerable increase in caspase-3 activity [96]. Sanhaji et al. reported that cell death was detected after treatment of tumor cells (BxPC3 and PANC-1) with conjugated MNPs with gemcitabine and NucAnt (N6L) and magnetic HT (43 °C, 1 h). The combination therapy generated arrest in the S phase of the cell cycle in PANC-1 cells, increased the intracellular ROS levels in BxPC-3 cells, and activated caspase-3, caspase-8, and JNK in BxPC-3 cells [97]. Lim et al. used a gemcitabine-loaded temperature-sensitive liposome in combination with 41 °C HT to treat colon adenocarcinoma cells CT26 and enhance the anticancer activity. The primary mechanism of cell death was a 1.5-fold increased caspase-3/7 activity and DNA fragmentation of chromatin [98].

### 3.12. Mapatumumab

Mapatumumab is a monoclonal antibody produced in humans that is now being evaluated in clinical trials for cancer treatment. The drug specifically targets TRAIL-R1, known as death receptor 4 (DR4), which is expressed on the surface of many different types of tumor cells [99].

Song et al. reported that the combination of HT and mapatumumab induced apoptosis in human colorectal carcinoma cell lines, CX-1 and HCT116. An increase in ROS production contributed to the synergetic effects that were mediated by activation of JNK and Bax oligomerization. It also produced the translocation of Bax to the mitochondria, loss of mitochondrial membrane potential, and cytochrome c release to the cytosol. Moreover, it resulted in caspase activation, and an increase in poly (ADP-ribose) polymerase cleavage [100]. In 2012, Song, Kim, and Lee reported that heat (42 °C, 1 h) and oxaliplatin sensitized colon cancer cell CX1 cells to mapatumumab via the mitochondrial-dependent apoptotic route and enhanced ROS generation, resulting in Bcl-xL phosphorylation at serine 62 in a JNK-dependent manner. Bcl-xL was detached from Bax during the multimodality treatment, allowing for Bax oligomerization [101]. Mapatumumab was combined with alkylating chemical oxaliplatin that reacted with DNA to generate intrastrand/interstrand DNA crosslinks. This intrastrand/interstrand crosslink impairs DNA base pairing, replication, and transcription, ultimately resulting in cell death [102]. In addition, Song et al. reported the synergistic effects of a co-treatment of oxaliplatin, mapatumumab, and HT (42 °C, 1 h) on apoptosis. The mechanism was caspase-dependent, activating both the extrinsic and intrinsic apoptotic pathways. In addition, the multi-modality therapy increased JNK signaling and reduced the amount of c-FLIPL in CX-1 and HCT116 cells [103].

### 3.13. Cyclophosphamide (CTX)

CTX is an alkylating agent that belongs to the oxazaphosporine class of drugs. There is considerable clinical experience in using CTX to treat cancer as an immunosuppressive agent to treat autoimmune and immune-mediated illnesses [104]. CTX exposure at low metronomic doses or HT can increase TSP-1 expression, an endogenous inhibitor of angiogenesis [105]. Borkamo et al. reported that TSP-1 expression was increased in the vascular matrix of microscopic arteries by a combination of low-dose CTX and HT. The vascular matrix immunostaining of TSP-1 was weak in tumors when treated with CTX alone, moderate in tumors treated only with HT, and significant in tumors when co-treated CTX and HT [106].

### 3.14. CTX and Melatonin

CTX is one of the most effective and commonly used anti-cancer agents. Its anti-cancer therapeutic value and immunosuppressive effects are amplified by differential aldehyde dehydrogenase expression in cells [107]. Lushnikova reported that in Walker 256 carcinosarcoma, a combination of whole-body heat with CTX and melatonin resulted in the greatest suppression of tumor growth with increased necrotic and apoptotic cell death [108].

### 3.15. Erlotinib

Erlotinib exhibited anti-cancer effects by inhibiting epidermal growth factor receptor (EGFR). Therefore, it inhibits cancer cell proliferation and blocks cancer cells from spreading throughout the body [109]. Zhang et al. used erlotinib-loaded MoS2 nanosheets functionalized with hyaluronic acid (MoS2-SS-HA-Er) as a tumor-targeting chemotherapeutic nanocarrier to integrate NIR photothermal with erlotinib. MoS2-SS-HA-Er combined with NIR increased the apoptosis rates of A549, PC-9, and H1975 cells significantly. The combinational therapy enhanced apoptosis in vitro and in vivo by arresting cancer cells in the G0/G1 phase [110].

### 3.16. Macrosphelide (MS5)

Ahmed et al. reported that HT at 41 °C for 20 min increased the apoptosis of human lymphoma U937 cells caused by MS5. The combination treatment of HT and MS5 resulted in a considerable increase in ROS generation, Fas externalization, and caspase-8/3 activation. The combination therapy also affected the expression of apoptosis-related proteins initiated by Bid cleavage and Bcl-2 downregulation [111]. 

### 3.17. Mafosfamide

Mafosfamide, an alkylating medication that can be used as a substitute for CTX, can cause apoptosis in mature human T-lymphocytes. Apoptotic cell death in lymphocytes was defined by characteristic morphological alterations, nucleosomal DNA fragmentation, and quantification of fragmented DNA using 3’-OH end labeling [112]. Ehlers et al. showed that the combination of mafosfamide and HT resulted in increased cellular damage. A combination therapy with mafosfamide at 10 M plus 1 h of 41.7 °C or 43 °C HT led to a significant increase in cell death of MSTO-211H human biphasic malignant pleural mesothelioma cells. HT enhanced the cytotoxicity of mafosfamide by generating DNA double-strand crosslinks and breakage, depleting nucleotides, such as NAD and ATP, and producing highly reactive and cytotoxic acrolein [113].

### 3.18. Melphalan (Mel)

In 2008, Krause, Kluttermann, and Mauz-Korholz reported that Mel has dose-dependent cytotoxicity in the Ewing’s sarcoma cell line RD-ES cells, which was enhanced by the concurrent administration of HT (42 °C, 2 h). Mel, HT, and their combination induced a substantial increase in caspase-3 activation. Both stimuli activated caspase-3, a major effector caspase in apoptosis [114].

### 3.19. Methotrexate (MTX)

MTX is an anti-cancer medication used to treat a variety of cancers [115]. MTX has become the first disease-modifying anti-rheumatic medicine in treating rheumatoid arthritis, psoriasis, and juvenile idiopathic arthritis. MTX, as a folate antagonist, affects the activity of the folate-dependent enzymes and purine and pyrimidine synthesis essential for DNA and RNA synthesis in rapidly proliferating malignant cells [116].

MTX alters cell cycle development by arresting cells in the S phase. Costa Lima et al. studied that together with mild HT (42 °C); more MTX was released from the superparamagnetic iron oxide nanoparticles (SPIONs)-loaded pegylated polymeric nanospheres than in normal temperature. The combination of HT, MTX, and SPIONs delivery suppressed the G1/G0 phase and cell cycle arrest in the S phase in Caco-2 and SW-480 cancer cells [117]. Li et al. also investigated the effects of MTX functionalized covalently onto iron-gold alloy magnetic nanoparticles (Fe-Au alloy nanoparticles, or NFAs). The efficacy of this drug-nanoparticle conjugate (NFA-MTX) on HepG2 (liver cancer) cell growth was studied. The heat was generated by NFAs, resulting in the cleavage of Au–S bonds and the release of MTX. The drug–nanoparticle combination was more readily absorbed by cancer cells (human hepatocellular carcinoma HepG2 cells) than normal cells (mouse fibroblast L929 cells), revealing its ability to target cancer cells specifically [118].

### 3.20. Mitomycin C (MMC)

MMC is commonly used as an intravesical cytotoxic agent [119]. DNA interstrands are induced through cross-links in the DNA molecules of the cells by MMC. Through this, MMC inhibits DNA replication and transcription [120].

Kim et al. reported that combination treatment with MMC and heat resulted in the synergistic induction of apoptosis in human colon cancer LS174T, LS180, HCT116, and CX-1 cells via the elevation of caspase activation. Furthermore, the increased JNK-Bcl-xL-Bak pathway was responsible for transmitting the synergistic action via the mitochondria-dependent apoptotic pathway [121]. van den Tempel et al. reported that the combined treatment of HT and cisplatin or MMC induced apoptosis by releasing cytochrome C and damaging homologous recombination DNA repair. In RT112 and T24 cells, the population of apoptotic cells increased by HT and cisplatin or MMC more than HT or chemotherapy alone. The RAD51 focus-formation assay detected morphological and numerical changes to the RAD51 foci upon HT in the RT112 and T24 cell lines [122].

### 3.21. Picibanil (OK-432)

OK-432 was developed in 1966 by a Japanese scientist Okamoto to treat lymphangioma. Although it is not painful during injection, an inflammatory reaction linked to edema, erythema, and a low-grade fever for up to five days may occur [123]. OK-432 exerts its anti-tumor activity directly by reducing tumor growth and indirectly by activating tissue inflammation, activating immune cells, and inducing cytokine production in the CT26 xenograft model. Li et al. studied the synergistic effect of OK-432 and pulsed-wave ultrasound HT (pUSHT). The combination group increased interferon gamma (IFNγ) positive CD4 T cells by 8.6-fold, IFNγ positive CD8 T cells by 6.7-fold, and NK cell infiltration by 4-fold. The immune response elicited by the combined treatment (pUSHT with local OK-432) slowed tumor growth. Moreover, increased systemic anti-tumor immunity could limit the growth of distal untreated tumors [124].

### 3.22. TRAIL

TRAIL is a member of the tumor necrosis factor group that activates the apoptosis pathway by activating its receptors, DR4 and DR5. Yoo and Lee reported that HT with TRAIL sensitized the effects of oxaliplatin and melphalan. TRAIL and heat (42 °C, 1 h) combination enhanced cytotoxicity in CX-1 cells compared to single treatments. In the presence of high concentrations (50 mg/mL) of oxaliplatin or melphalan, HT increased TRAIL-induced caspase-3/8 activation. Furthermore, DNA fragmentation and enhanced cytochrome-c release at elevated temperatures induced apoptotic cell death [125].

### 3.23. Pelitinib

Pelitinib is a powerful irreversible epidermal growth factor receptor (EGFR) tyrosine kinase inhibitor studied in clinical studies to treat lung cancer. Pelitinib was initially noted for its potential to reverse MDR in cancer cell line models overexpressing the three major multidrug resistance (MDR) transporters (ABCB1/P-gp, ABCC1, and ABCG2). After heat exposure, only the combination of topotecan and pelitinib induced significant apoptosis. In particular, pelitinib inhibited the up-regulated ABCB1 and ABCG2 in response to HT, thereby potentiating apoptosis [126].

### 3.24. Pluronics

Pluronics (known as poloxamers) are a family of triblock copolymers composed of hydrophilic ethylene oxide (EO) and hydrophobic propylene oxide (PO) with the general structure EOx-Poy-Eox. They have been incorporated into many recent therapeutic formulations for cancer treatment. Pluronics with a molecular weight range of 1100–3200 Da and hydrophilic–lipophilic balance < 8 had the highest ability of thermosensitizing capacity. Pluronics with L31, L61, L62, L10, and L64 had the maximum benefit for HT. Krupka and Exner reported that L61, in combination with HT, resulted in a dramatic increase in caspase-3/7 activity 24 h after treatment. In vivo studies involving DHD/K12/TRb rat colorectal adenocarcinoma cells indicated that tumors treated with a combination of L61 pretreatment and RF ablation treatment significantly decreased the tumor size compared to tumors treated with RF ablation alone at three and four weeks after treatment [127].

### 3.25. Ranpirnase

Ranpirnase, a cytotoxic amphibian ribonuclease, has an anti-cancer effect by triggering apoptosis in the absence of the p53 protein [128]. Ranpirnase causes apoptosis in multiple myeloma cells in vitro and in vivo, inhibiting tumor growth in xenografted severe combined immunodeficiency (SCID) mice by disrupting the NF-κB pathway and MMP9 activity [129]. Halicka reported that the cytotoxicity and overall pro-apoptotic effect was greater in the co-treatment of ranpirnase and 41 °C-HT than ranpirnase or HT alone. The treatment of 2 or 5 g/mL ranpirnase at 40 °C for 24 or 48 h increased the incidence of apoptosis by 64 to 200%. Apoptosis was determined by the frequency of cells expressing activated caspase-3 or activated serine proteases in human lymphoblastoid TK6 cells [130].

### 3.26. Sorafenib (SRF)

SRF is an orally administered multi-kinase inhibitor that inhibits a variety of cell surface tyrosine kinases, including the VEGF receptor (VEGFR)-1, VEGFR-2, VEGFR-3, platelet-derived growth factor receptor (PDGFR)-, KIT, FLT-3, RET, and RET/PTC [131]. Wu et al. developed SRF/ICG nanoparticles (SINP) that can rapidly infiltrate the HCC cell line Huh7 and its xenograft tumor cells to achieve potent cytotoxicity in response to NIR laser irradiation. After laser irradiation, the SINP-treated group showed significantly higher ROS generation and cleavage of PARP and caspase-3/9 [132].

### 3.27. SurvivinT34A

A threonine 34 to alanine mutation of survivin (survivinT-T34A) abolished the phosphorylation site for p34(cdc2)–cyclin B1 initiating the mitochondrial apoptotic pathway in cancer cells [133]. Li et al. reported that survivinT34A alone could hinder the anti-apoptotic activities of HSP90 and survivin. Moreover, HT induced the accumulation of survivinT34A in CT26, B16-F10, and MethA cells. HT also activated numerous apoptotic pathways, including p53, JNK, and other p53-independent mechanisms. Furthermore, suppressing survivin in conjunction with heat helped prevent tumor-associated angiogenesis in the in vivo model of CT26 xenograft [134].

### 3.28. Temozolomide

Temozolomide, a DNA-methylating agent that triggers apoptotic cell death, is commonly prescribed to treat malignant glioma. Temozolomide induces senescence and the downregulation of the DNA repair mechanisms in glioma cells [135]. Using rats bearing human melanoma DM6 xenografts, Ko et al. reported that a significantly larger proportion of apoptotic cells is observed in cells treated with temozolomide when combined with moderate heat at 41 °C [136].

### 3.29. Tirapazamine (TPZ)

TPZ is the second anti-cancer medication that acts predominantly as a hypoxia-selective cytotoxin. Masunaga et al. reported that mild temperature HT (40 °C, 1 h) sensitized human head and neck squamous cell carcinoma cell line SAS cells to TPZ. In addition, the effects of paclitaxel, gamma-ray irradiation, and cisplatin were all enhanced significantly when HT and TPZ were combined with each therapy. The combination with mild temperature HT and TPZ induced cell cytotoxicity, with a mechanism independent of the p53 status [137].

### 3.30. Toremifene

Toremifene is a chlorinated tamoxifen analog used to treat hormone-dependent breast cancer in postmenopausal women. Toremifene inhibits cell growth by inhibiting mitosis and promoting apoptosis. The activation of TGF-β1 and suppression of IGF-1 may be involved in these occurrences [138]. Kanaya et al. reported that the combined therapy groups with toremifene and local HT (43.5 °C, 30 min) had significantly stronger anti-tumor effects against MCF-7 xenografts than the single therapy groups. The combined therapy groups had lower estrogen receptor expression and more G0/G1-phase cells and fewer S-phase cells than groups with sole treatments. Furthermore, the combination group showed a higher apoptotic index and lower tumor vascular density [139].

### 3.31. Trabectedin

Trabectedin is a tetrahydroisoquinoline molecule that binds with the N2 of guanine in the minor groove of DNA, causing DNA damage and changing the transcription regulation in the promoters and genes. Regardless of the p53 status, this drug increases cell cycle arrest and death in cancer cells. The impact was amplified in cells lacking a homologous recombination [140]. Hyperthermic temperatures (41.8 or 43 °C) increased clonogenic cell death and G2/M cell cycle arrest dramatically in various types of human sarcoma cell lines and colorectal carcinoma cell lines. The combination of HT and trabectedin resulted in a significant increase in the accumulation of γH2AX foci, a critical marker of double-strand breaks (DSBs). HT increased the efficiency of trabectedin in BRCA2-positive cells, but the effect was lower in BRCA2-deleted or siRNA-transfected BRCA2 knockdown cells [141].

### 3.32. Vinblastine

Vinblastine, an alkaloid derived from *Catharanthus roseus* (L.) G. Don, is a widely utilized anticancer drug in clinical practice [142]. The drug suppresses microtubule polymerization by selective binding to tubulin at the vinca alkaloid-binding sites [143]. Vinblastine is an efficient inhibitor of angiogenesis when used in low dosages, but at higher doses, it destroys the tumor blood vessels and induces hemorrhagic necrosis in solid mouse tumors. HT also exhibits anti-vascular activity in experimental tumor models. Eikesdal, Bjerkvig, and Dahl reported that the combined treatment of HT and vinblastine resulted in significant vascular injury and the extensive development of hemorrhagic necrosis in the tumor parenchyma [144].

**Table 2 antioxidants-11-00625-t002:** **2-1.** Sensitizing effect of HT on cisplatin. **2-2.** Sensitizing effect of HT on Paclitaxel. **2-3.** HT sensitized tumor cells to various anti-cancer agents 1. **2-4.** HT sensitized tumor cells to various anti-cancer agents 2.

**2.1.**				
**Compound**	**HT**	**Cell Line/Observation Model**	**Molecular Mechanism**	**Ref**
Cisplatin	43 °C, 1 h	Prostate cancer, PC-3, DU-145/in vitro	cleavage of caspase-3/activation of AMPKα-JNK and inhibition of Akt-mTOR-p70s6k signaling pathway	[48]
Cisplatin	44 °C, 30 min	Human maxillary squamous cell carcinoma, cisplatin-resistant, IMC-3, IMC-3CR/in vitro	Increase of ROS production, repression of SESN1	[49]
Cisplatin	42–43 °C, 30 min	Ovarian cancer, SKOV3/in vitro, in vivo	Apoptosis	[50]
Cisplatin	42.5 °C, 1 h	NSLCLs, A549, H460/in vitro	Pharmacokinetics change	[51]
Cisplatin	44 °C, 30 min	Human maxillary carcinoma, IMC-3/in vitro	Cleavage of PARP	[52]
Carboplatin (CPt)	42.5 °C, 1 h	Human retinoblastoma, WERI/in vitro	Cell cycle arrest (G2/M)/caspase 9 activation induced by the release of cytochrome C	[54]
Cisplatin and 5-fluorouracil	42 °C	Orthotopic esophageal squamous cancer/in vitro, in vivo	Apoptosis	[56]
Cisplatin and doxorubicin	42 °C, 1 h	Human Ovarian Carcinoma, OVCAR8/Colon cancer cell, HCT116/in vitro	DNA damage (blocking PARylation)	[58]
Cisplatin ferucarbotran (Resovist)	42.5 °C, 20 min	Human oral cancer, HSC-3, OSC-19/in vitro	Cell cycle arrest (G2/M)	[60]
Sodium arsenite (NaAsO_2_) and cisplatin	39 °C, 1 h	Ovarian cancer, A2780, CP70/in vitro	Cell cycle arrest (G2/M)	[62]
Paclitaxel and cisplatin	43 °C, 1 h	Osteosarcoma, OS732, MG63/in vitro	Apoptosis (up-regulating Fas)	[63]
**2-2.**				
**Compound**	**HT**	**Cell Line/Observation Model**	**Molecular Mechanism**	**Ref**
Paclitaxel	41 °C, 2 h	Human breast cancer, MCF-7/in vitro	Apoptosis	[65]
Paclitaxel	41.5, 43 °C, 2 h	Human breast, MCF-7/Ovarian cancer, SKOV-3/Hepatocellular carcinoma, HepG2/in vitro	Cell cycle arrest (G2/M)	[66]
Paclitaxel	41 °C, 2 h	Human Breast cancer, MCF-7/in vitro	activation of caspase-7/Cell cycle arrest (G2/M)	[67]
Paclitaxel (PTX)	42 °C, 2 h	Human larynx carcinoma, HEp2/in vitro	Cell cycle arrest (G2/M)	[68]
Taxol	41.5 °C, 2 h	Cervical adenocarcinoma, HeLa/in vitro	Necrosis/Cell cycle arrest (G2/M)	[69]
Docetaxel	41 °C, 2 h	Human breast cancer cell line MCF-7 and MDA-MB-453/in vitro	Down-regulation of proteins in the Bcl-2 family	[71]
**2-3.**				
**Compound**	**HT**	**Cell Line and Observation Model**	**Molecular Mechanism**	**Ref**
Doxorubicin and GSH	42 °C, 1 h	NSLCLs, A549/in vitro	Increase of the GSH-mediated release of doxorubicin	[73]
Doxorubicin	42 °C	Squamous cell carcinoma, SCC-7/in vivo	Apoptosis	[74]
Doxorubicin	42 °C, 2 h	Murine colorectal adenocarcinoma, C26/in vitro	upregulation of p53	[75]
Doxorubicin	gold nanoshells 43 °C	Human hepatoma, BEL-7402/in vitro	MMP depolarization/DNA cross-linking and inhibition of DNA repair mechanisms	[76]
Doxorubicin	42 °C	Human hepatoma, Hep3B/in vivo	Apoptosis, Necrosis	[77]
Doxorubicin	43 °C, 1 h	Uterine cancer, MES-SA/in vitro	Apoptosis (overcoming P-gp regulated multidrug resistance)	[78]
Doxorubicin, IR-780 loaded polymeric prodrug micelles	IR-780, NIR imaging	Breast cancer, MCF-7, MCF-7, ADR/in vitro, in vivo	MMP depolarization	[79]
A water-responsive phospholipid-calcium-carbonate hybrid nanoparticle (PL/ACC-DOX&ICG)	43 °C, 1 h, with PL/ACC-DOX&ICG upon NIR laser irradiation.	Breast Cancer, 4T1/in vivo, in vitro	Apoptosis	[80]
Pegylated liposomal doxorubicin	Focused ultrasound system	Murine breast cancer, 4T1-luc2/in vivo	Apoptosis (increased perfusion, vascular permeability and interstitial microconvection)	[82]
Liposomal doxorubicin	42 °C, 30 min	Hepatocellular carcinoma, HepG2/in vivo, in vitro	MMP depolarization	[83]
BCM- doxorubicin (a block copolymer micelle (BCM) formulation (which may reduce toxicities of doxorubicin in a similar way to pegylated liposomal doxorubicin)	42 °C, 1 h	Human ovarian cancer HEYA8, OV-90, SKOV3/in vitro, in vivo	Growth inhibition (Pharmacokinetics change)	[84]
Lipodox^®^ (liposome-encapsulated doxorubicin)	42 °C, 30 min	Murine colon carcinoma, CT26/in vivo, in vitro	enhancement of the uptake of liposomal drugs by enhancing phagocytic activity	[86]
Redox-responsive hyaluronic acid nanogels	the laser irradiation at 808 nm, 60 s	Human breast cancer, MCF-7, ADR/in vitro	Apoptosis (increased intracellular doxorubicin accumulation)	[87]
**2-4.**				
**Compound**	**HT**	**Cell Line and Observation Model**	**Molecular Mechanism**	**Ref**
Bortezomib	44 °C, 30 min	Mantle cell lymphoma, Jeko-1, Rec-1, Granta 519, HBL-2, NCEB-1/in vitro	inhibition of HSP27/70	[90]
Bortezomib	44 °C, 15 min	Human leukemic monocyte lymphoma, U937/in vitro	Apoptosis (increase cells underwent the late apoptosis stage)	[91]
Gemcitabine	43 °C, 1 h	Human pancreatic carcinoma, AsPC-1, MIAPaCa-2/in vitro	Apoptosis, Necrosis (blocking the activation of NF-B)	[94]
Gemcitabine	42 °C, 90 min	Pancreatic cancer, SW1990/in vitro	activation of ROS/JNK signaling pathway/Cell cycle arrest (S-phase)	[95]
Gemcitabine	43 °C, 3 h	Human non-small-cell lung cancer, BZR-T33/in vitro	Cell cycle arrest (G2/M)/cleavage of caspase-3	[96]
Gemcitabine and NucAnt	43 °C, 1 h	Human pancreatic cancer, BxPC3, PANC-1/in vitro	Cell cycle arrest (S-phase)/activation of expression of p53, p21, BcL-2, PARP, Bax and H2AX	[97]
Gemcitabine-loaded TSL	41 °C, 24 h	Adenocarcinoma, CT-26/in vivo, in vitro	cleavage of the caspse-3/7 and causing the fragmentation of chromatin DNA	[98]
Mapatumumab	42 °C, 1 h	Human colorectal carcinoma, CX-1, HCT116/in vitro	Elevation of ROS level/JNK activation/MMP depolarization	[100]
Mapatumumab and Oxaliplatin	37 °C, 3 h	Human Colon Cancer, CX-1/in vitro, in vivo	MMP depolarization/ROS production	[101]
Mapatumumab and Oxaliplatin	42 °C, 1 h	Human colorectal carcinoma, CX-1, HCT116/in vitro	Activating JNK signaling pathway	[103]
Cyclophosphamide	43 °C, 1 h	Glioblastoma-like tumor, BT4An/in vivo	Anti-angiogenisis (upregulation of TSP-1)	[106]
Cyclophosphamide and melatonin	43.5 °C	Carcinosarcoma, Walker 256/in vivo	N/A	[108]
Erlotinib	NIR irradiation, 50 °C within 500 s	NSLCLs, A549, H1975/Prostate cancer, PC-9/in vitro, in vivo	Cell cycle arrest (G0/G1)/inhibiting the epidermal growth factor receptor (EGFR) tyrosine kinase	[110]
Macrosphelide	41 °C, 20 min	Human lymphoma, U937/in vitro	increasing in ROS generation and caspase 3, 8 activation/down-regulation of Bcl-2	[111]
Mafosfamide	41.7 °C, 1 h	Pleural mesothelioma, MSTO-211H/in vitro	Necrosis (shifting cell death from apoptosis to necrosis)	[113]
Melphalan	42 °C, 2 h	Human Ewing tumor, RDES/in vitro	activation of caspase-3	[114]
Methotrexate	mild HT (42 °C)	Human hepatoma, Caco-2, SW480/in vitro	Cell cycle arrest (S phase)	[117]
Methotrexate	40 °C–50 °C, 1 h	Hepatocellular carcinoma, HepG2/in vitro	enhancement of the uptake of liposomal drugs	[118]
Mitomycin C	42 °C, 1 h	Human colon cancer, LS174T, LS180, HCT116, CX-1 cell/in vitro	Activation of JNK pathway induced mitochondria-dependent apoptotic pathway	[121]
Mitomycin C	42 °C, 12 min	Bladder cancer cell, RT112 and T24/ex vivo	cytochrome C release/HR DNA damage repair capacity decrease	[122]
OK-432	(pUST)	Colorectal adenocarcinoma, CT26-luc tumor cell/in vivo	Apoptosis, Necrosis(tissue inflammation induced necrosis)	[124]
Oxaliplatin and Melphalan	42 °C, 1 h	Human colorectal cancer, CX-1/in vitro	MMP depolarization	[125]
Pelitinib	42.5 °C, 4 h	NSLCLs, A549 cell/in vitro	up-regulation of ABCB1/ABCG2	[126]
Pluronic L61	43 °C, 20 min	Rat colorectal adenocarcinoma, DHD, K12, TRb/in vivo	MMP depolarization	[127]
Ranpirnase	40 °C, 24 h	Human lymphoblastoid, TK6/in vivo, in vitro	Apoptosis	[130]
Sorafenib, indocyanine	785 nm irradiation for 10 min at 2 W/cm^2^	Hepatocellular carcinoma, Huh7/in vitro, in vivo	producing ROS and activating caspase-9, 3	[132]
SurvivinT34A	42 °C, 1 h	Murine colorectal carcinoma, CT26/Murine melanoma, B16-F10, MethA/in vitro, in vivo	activation of p53/bound to Hsp90 and abrogating the cytoprotection of Hsp90	[134]
Temozolomide	43 °C, 15 min	Human melanoma, DM6/in vitro, in vivo	N/A	[136]
Tirapazamine	40 °C, 1 h	human head and neck squamous cell carcinoma, SAS/in vitro	p53 independent apoptosis	[137]
Toremifene	43.5 °C, 30 min	Breast cancer, MCF-7/in vitro	weakening Estrogen receptor expression/G0/G1-phase cells↑ and S-phase cells↓	[139]
Trabectedin	41.8 °C and 43 °C, 90 min	Human sarcoma cell osteosarcoma, U2OS/Liposarcoma, SW872/Synovial sarcoma, SW982/Ewing sarcoma, RD-ES/Leiomyosarcoma, SKUT-1/Human colorectal carcinoma, DLD1/in vitro	BRCA2 degradation and impairment of DNA homologous recombination repair	[141]
Vinblastine	44 °C, 1 h	BT4An rat glioma/in vivo	Apoptosis (disturbing established neovasculature, and producing vascular shutdown)	[144]

## 4. Natural Products

Several natural products are used for cancer therapy and are effective. Their main advantages are reasonable prices, fewer side effects, and the ability to interact with multiple targets simultaneously. Despite these advantages, the apoptotic effect of some natural products is difficult to maintain when used alone. Studies have shown that combined treatment of HT and natural products may enhance and correctly deliver the apoptotic effects more efficiently. This section and Table 3 summarized and categorized the effects of combined treatments with HT and natural products according to the action mechanisms.

### 4.1. Reactive Oxygen Species Production

#### 4.1.1. Baicalin

Baicalin is a compound series extracted from a traditional Chinese herb, *Scutellaria baicalensis*. It generates ROS and induces an apoptosis [145]. The combination of HT at 44 °C for 12 min and baicalin induces the loss of the mitochondrial membrane potential, increases oxidative stress, and suppresses the antioxidant system. The up-regulation of ER stress and autophagic markers followed by increased apoptosis were observed in the myelomonocytic leukemia cell line U937 [146]. 

#### 4.1.2. Epigallocatechin Gallate (EGCG) and Chlorogenic Acid (CGA)

EGCG is a major flavonoid in green tea. CGA is a major phenolic acid in coffee [147]. Thermal cycling is a method to gain an equivalent thermal dosage through repeated heat-and-cold cycles. Lu et al. demonstrated this strategy of thermal cycling-HT at 36–43.5 °C with EGCG or CGA in human pancreatic cancer cells, PANC-1. The result showed arrest at the G2/M phase and the induction of the ROS-dependent mitochondria-mediated apoptosis. The synergism shown was attributed to the thermal damage caused by HT [148].

#### 4.1.3. Cinnamaldehyde (CNM) 

CNM is the main biologically active compound extracted from the essential oil of Cortex cinnamon. CNM works via a TLR4-dependent signaling pathway leading to apoptosis [149]. The combination of CNM and HT at 43 °C for 30 min promoted apoptosis in the NSCLC cell line, A549. The pro-apoptotic effect of a combination treatment was dependent on the increase in ROS and MAPKs, which are the downstream targets of ROS [150]. Another in vitro study reported that the combination of CNM and HT increased cytotoxicity in the renal cell carcinoma cell line, ACHN cells, by inhibiting HSP70 expression, arresting the cell cycle, and increasing ROS generation [151]. 

#### 4.1.4. Nonivamide

Nonivamide is a pungent composition similar to capsaicin. The slight difference with capsaicin comes from one methyl group on the carbon chain and one double bond. Lu et al. used U937 cells to evaluate the effects of combined therapy of HT and nonivamide. First, U937 cells were re-treated with 50 μM of nonivamide and incubated at 44 °C for 15 min. The cells were then treated for 3 h after the HT treatment. The results showed a spike in ROS in the cells resulting in mitochondrial dysfunction and p38 and JNK-mediated apoptosis [152].

#### 4.1.5. 5Z-7-Oxozeaenol (OZ)

OZ irreversibly inhibits transforming growth factor-β activated kinase 1 (TAK1), a factor that can achieve intrinsic ATPase activity [153]. The drug can inhibit the NF-κB pathway and produce ROS, cleaving caspase-3 and 7 in HT-29 and HeLa cancer cells [154]. Peng et al. studied the combined effect of HT and OZ in an in vitro study using human T lymphoblast Molt-4 cells. OZ treatment-induced cancer apoptosis by inhibiting HSP70, increasing p38 and JNK, and inducing ER stress. On the other hand, the combination of OZ and HT at 44 °C for 10 min enhanced apoptosis compared to using OZ alone [155].

#### 4.1.6. Withaferin A (WA)

WA is found in an Asian medicinal plant (*Withania somnifera*) and is regarded as a potential anti-cancer steroidal lactone. The Ehrlich ascites tumor cell study from the early 70 s provided the first evidence that WA can inhibit cancer proliferation [156]. Cui et al. reported that JNK phosphorylation was induced by the combination of WA and HT at 44 °C for 30 min, and ERK phosphorylation was suppressed in HeLa cells. In addition, this combination treatment depolarized MMP and induced the cleavage of caspase-3 [157].

#### 4.1.7. Ascorbic Acid (AscH2) 

AscH2 (also known as vitamin C) is a water-soluble ketolactone with two water-soluble hydroxyl groups. The cytotoxicity of high doses of AscH2 in cancer cells is related to ROS production, which reduces energy synthesis via glycolysis [158]. APPS (the trisodium salt of ascorbic acid-2-phosphate6-O-palmitate), A6-P, APHD (an isomer of APPS), and VCIP are AscH2 derivatives. When combined with HT, these derivatives exerted a significant carcinostatic effect. Saitoh et al. reported that APPS outperformed APHD, A6-P and VCIP in terms of anti-cancer effects at normal temperature (37 °C) and had synergistic effects on carcinogenesis at 42 °C in human tongue squamous carcinoma HSC-4 cells, accompanied by reduced cytotoxicity in normal cells (human dermal fibroblasts OUMS-36). This study shows that the combined treatment of AscH2 derivatives and HT induces apoptosis without side effects [159]. 

#### 4.1.8. Docosahexaenoic Acid (DHA)

DHA can decrease the enzymatic activity of PTP1B phosphatase and the viability of MCF-7 breast cancer cells. DHA is a potent anti-cancer agent [160]. DHA consists of 22 carbons, six double bonds, and unsaturated fatty acids [161]. By co-treatment with 44 °C for 10 min, apoptosis of U937 cells increased, with ROS generation, the phosphorylation of protein kinase C (PKC)-d, and mitochondrial dysfunction at higher levels than DHA alone. Such synergistic effects may be due to the 22 carbons and six double bonds of DHA because the combined DHA–HT treatment showed increased ROS generation, which was found 3 h after the experiment [162].

### 4.2. Regulation of Anti/Pro-Apoptotic Transcription Protein 

#### 4.2.1. ch282-5

BH3-only proteins (BIM, PUMA, BID, BAD, BIK, BMF, NOXA, and HRK), in favor of apoptosis, share BH3 domains. BH3-mimetics activates apoptosis directly by binding and selectively inhibiting anti-apoptotic BCL-2 family members, thereby bypassing the requirement for upstream initiators, such as p53 [163]. Ch282-5 is a gossypol (isolated from *Gossypium*) derivative acting as a BH3-mimetic. Combined HT at 43 °C for 1 h and ch282-5 increased the ratio of apoptotic cells in human and murine melanoma cell lines by suppressing anti-apoptotic proteins, such as Bcl-2 and IAP family, disturbing the mTOR/p70S6k signaling pathway while stimulating JNK and p38 [164].

#### 4.2.2. Crocin

Crocin is a water-soluble carotenoid pigment, and the stigmas of *Crocus sativus* L. are its natural source. Crocin interacts with various cellular proteins, such as structural proteins, membrane transporters, and enzymes, which play essential roles in ATP synthesis, redox homeostasis, and signal transduction [165,166]. Crocin and HT at 43 °C for 2 h increased Bax expression while simultaneously decreasing Bcl2. In addition, HSP70 and 90 were reduced by the combination treatment. MDA-MB-468 cells treated with crocin (1.5 and 3 mg/mL) combined with HT showed remarkably suppressed colony formation and LDH release [167]. 

#### 4.2.3. Perillyl Alcohol (POH)

POH, which is extracted from the essential oils of lavender, peppermint, and other plants, is a natural dietary monoterpene. As a metabolite of limonene, POH is derived from the mevalonate/isoprenoid pathway. Although the molecular and cellular impact of POH is often identified as “a Ras inhibitor”, its effect can come in various forms by regulating multiple cellular targets and growth-regulatory processes [168,169]. POH and HT (43 °C for 1 h) induced a synergistically enhanced apoptotic effect in SCK mammary carcinoma cells. The combined treatment induced the TGF-β–dependent signaling pathways and subsequent G1 arrest. Pro-apoptotic factors, such as Bax, Bak and Bad, were also increased [170].

#### 4.2.4. 5-Aminolevulinic Acid (5-ALA)-Mediated PDT

5-ALA is a natural amino acid found in animals and plants, a common precursor of hemoglobin and chlorophyll. Through various steps involving several precursors in the cytoplasm, 5-ALA leads to the biosynthesis of PpIX, the last precursor of heme, in mitochondria [171]. Hirschberg et al. subjected 5-ALA photodynamic therapy or HT, and the combination of the two to human or rat glioma spheroids. At temperatures below 49 °C, HT did not influence spheroid survival, whereas sub-threshold 5-ALA photodynamic therapy caused a slight decrease in survival. On the other hand, when both HT (40–46 °C) and 5-ALA photodynamic therapy were given simultaneously, they showed a synergistic anti-survival effect in glioma spheroids [172]. 

#### 4.2.5. Curcumin

Curcumin, a polyphenol extracted from *Curcuma longa*, has anti-cancer effects through various cell-signaling pathways, such as growth factors, cytokines, transcription factors, and apoptosis [173]. Combined treatment with curcumin and HT (42 °C, 1 h) in murine Lewis lung (LL) carcinoma cell line and endothelial cells induced apoptosis. In the LL/2 in vivo model, the combination prevented cancer growth. Furthermore, the co-treatment reduced angiogenesis and increased the apoptosis of cancer [174].

### 4.3. Mitochondrial Membrane Potential (MMP) Depolarization

#### 4.3.1. Furan-Fused Tetracyclic Synthesized Compounds (DFs) 

A chain of furan-fused tetracyclic analogs has been used to treat cancer. DFs are furan-fused tetracyclic analogs. DFs were first synthesized when the anti-viral activity of a series of related derivatives was investigated. Despite DF3 alone not initiating abundant ROS to apoptosis, the combined therapy with HT at 44 °C for 20 min activated caspase-3/8 and decreased the mitochondrial transmembrane potential. As a result, it released cytochrome c to the cytosol and the expression of Fas and subsequent apoptosis was induced in human lymphoma U937 cells [175].

#### 4.3.2. Betulinic Acid (BA)

BA is a lupane-type pentacyclic triterpenoid saponin recognized for its inhibitory effect on different kinds of malignancies. Its anti-cancer effect is by triggering apoptosis in the mitochondria-related pathway and regulating the cell cycle [176]. The combination of HT at 42 °C for 2 h and BA showed increased apoptosis in human melanoma DB-1 cells. BA and the mitochondrial membrane adhered irreversibly at pore transition complex sites. Moreover, at low pH-adapted cells, BA + HT generated even greater levels of apoptosis and cytotoxicity than at normal pH (pH 7.3) due to the higher drug uptake [177].

#### 4.3.3. Curcumin and 5-FU

5-FU is a fluoropyrimidine analog that inhibits thymidylate synthase (TS) and DNA synthesis/repair in cancer cells [178]. A previous study combined curcumin and 5-FU with magnetic nanoparticles encapsulating poly (D,L-lactic-co-glycolic acid) nanoparticles to achieve synergy. With magnetic HT, the drugs were delivered in a short period. Hence, the cytoskeletons and mitochondrial membrane potential were disrupted, and apoptosis was induced in human breast adenocarcinoma cells and human glial cells [179]. 

### 4.4. Cell Cycle Arrest

#### Arsenic Trioxide (As_2_O_3_)

As_2_O_3_, a component of the traditional Chinese medicine arsenic, possesses anti-cancerous characteristics [180]. As₂O₃ produces ROS, which induce cell death in leukemia and other solid tumors (lung cancer, breast cancer, prostate cancer, gastric cancer, cervical cancer, bladder cancer, pancreatic cancer, nasopharyngeal cancer and ovarian cancer) [181]. A combination of As_2_O_3_, HT, and radiotherapy synergistically arrested human myelomonocytic lymphoma U937 cells at the G_2_/M phase. Consequently, apoptosis was increased by this combined therapy [182].

### 4.5. Regulation of Heat Shock Response

#### 4.5.1. Curcumin and Resveratrol

Resveratrol is a type of stilbene that occurs naturally in various plant-based materials, including berries, red wine, grapes, and other fruits and vegetables. Kuo et al. treated curcumin and resveratrol in combination with mEHT (42 °C, 30 min) to the murine colon cancer cell line CT26 to determine if apoptosis was induced and cell proliferation was repressed. A decrease in HSP70 expression was also observed. Furthermore, CD3+ T cells and F4/80+ macrophages were accumulated in the tumor tissue [183].

#### 4.5.2. Quercetin

Asea et al. evaluated quercetin using PC3 cells and found that a co-treatment with HT (43 °C, 1 h) significantly boosted the anti-cancer effect of quercetin. The combined therapy significantly slowed tumor growth. When treated with HT, quercetin prevented the development of DU-145. Quercetin inhibited the production of HSP72 in prostate cancer, suggesting that it may operate as an apoptosis sensitizer and tumor growth inhibitor in vivo [184]. Shen et al. found that quercetin suppressed the increased expression of HSP70 and P-gp in human myelogenous leukemia cell line K562/A in response to HT. A pre-treatment with quercetin resulted in an 8.3-fold increase in doxorubicin accumulation compared to the control cells. Furthermore, quercetin dose-dependently induced apoptosis and G2/M arrest, indicating that combining quercetin with HT resulted in synergism on apoptosis [185].

#### 4.5.3. Quercetin and Tamoxifen

The flavonoid quercetin inhibits the heat-induced production of HSPs in various cell types. In breast cancer cells, the effect of quercetin is cell-specific and may involve the inhibition of HSF transcriptional activity rather than DNA binding activity [186]. 

Tamoxifen is an anti-estrogen that inhibits the activity of estradiol in specific organs by acting as a competitive inhibitor. Tamoxifen resistance occurs when the estrogen receptor or other transcription factors are lost or altered [187]. The inhibitory effects of tamoxifen on cell growth and its cytotoxicity are closely linked to protein kinase C [188]. Concomitantly with Qin et al. [40], quercetin and tamoxifen can decrease heat-induced HSP70 in melanoma cells. Based on these properties, Piantelli et al. showed that co-treatment of HT at 42.5 °C, 1 h with quercetin and tamoxifen synergistically induce apoptosis in human melanoma cells M10, M14, and MNT1 [189].

#### 4.5.4. Quercetin + Lipopolysaccharide (LPS)

Zhou et al. used tumor-targeting agent AG conjugated with IR820 for tumor imaging and HT. Quercetin (an inhibitor of HSP70) and LPS (an activator of TLR-4) were also used as a combined treatment with HT. The combination therapy of quercetin, LPS, and HT inhibited tumor growth of human thyroid cancer TT cells both significantly more than the individual treatments, both in vivo and in vitro. Furthermore, AG-IR 820 modification increased the cellular absorption of quercetin and enhanced its ability to target tumors [190].

### 4.6. Other Mechanisms

#### 4.6.1. β-Lapachone (β-Lap)

β-lap, a quinine-containing compound obtained originally from lapacho trees in South America, has a considerable anti-tumor effect in many cancer species. The anti-tumor effect of β-lap is related to the activation of NAD(P)H:quinone oxidoreductase (NQO1). The combination treatment of β-lap and HT at 42 °C was positively more effective in HOS cancer cells. This effect can be achieved due to the increase in NQO1 activity by heat stimulation [191].

#### 4.6.2. Enediyne

Anti-tumor antibiotics based on enediyne are a type of DNA-damaging medication that is highly sensitive to changes in the DNA structure and dynamics. DNA DSBs can activate the response proteins of DNA damage, thereby inhibiting cell cycle progression and eventually resulting in apoptotic cell death. These processes may be the primary mechanism through which enediyne performs its anticancer action [192]. In 2020, Garrett et al. reported the clonogenic survival of U-1 melanoma or MDA-231 breast cancer cells when treated with varying amounts of copper, iron, and zinc complexes. In this study, HT at 42.5 °C was found to enhance apoptosis. HT accelerated the occurrence of DNA DSBs caused by these metal chemicals and impeded cell repair [193].

**Table 3 antioxidants-11-00625-t003:** **3-1.** HT + natural products (ROS production). **3-2.** HT + natural products (Regulation of anti/pro-apoptotic transcription protein). **3-3.** HT + natural products (MMP depolarization, Cell cycle arrest, Regulation of heat shock response).

**3-1.**				
**Compound**	**HT**	**Cell Line and Observation Model**	**Molecular Mechanism**	**Ref**
Baicalin	44 °C, 12 min	Myelomonocytic leukemia, U937/in vitro	Caspase activation/Bax and Noxa↑, Downregulation of antiapoptotic proteins/Bcl-2↓, MMP depolarization, increase of ROS, ER stress	[146]
Epigallocatechin gallate and chlorogenic acid	10-cycles at 43.5–36 °C	Human pancreatic cancer, PANC-1/in vitro	Cell cycle arrest (G2/M)/the induction of the ROS-dependent mitochondria-mediated apoptosis	[148]
Cinnamaldehyde	43 °C, 30 min	NSLCLs, A549/in vitro	ROS production and Mitogen-Activated Protein Kinase Family↑	[150]
Cinnamaldehyde	43 °C, 30 min	Renal adenocarcinoma, ACHN/in vitro	inhibition in HSP70 expression, Cell cycle arrest, increase of ROS	[151]
Nonivamide	44 °C, 15 min	Human lymphoma, U937/in vitro	elevation of intracellular ROS/mitochondrial dysfunction/increased activation of JNK and p38	[152]
5Z-7-oxozeaenol	44 °C, 10 min	Human T lymphoblast, Molt-4/in vitro	HSP70↓/p38and jnk↑/ROS production (ER stress-induced apoptosis)	[155]
Withaferin A	44 °C, 30 min	Human cervical cancer, HeLa/in vitro	inducing JNK phosphorylation (p-JNK), and decreases in the phosphorylation of ERK (p-ERK)	[157]
Ascorbic acid	42 °C, 15 min	Ehrlich ascites tumor, EAT/in vitro	Cell cycle arrest (G2/M)/H_2_O_2_ induced apoptosis	[159]
Docosahexaenoic acid	44 °C, 10 min	Human myelomonocytic lymphoma, U937/in vitro	MMP depolarization (inducing phosphorylation of protein kinase C (PKC)-d)	[162]
**3-2.**				
**Compound**	**HT**	**Cell Line and Observation Model**	**Molecular Mechanism**	**Ref**
Ch282-5	43 °C, 1 h	Melanoma, M21, B16F10/in vitro, in vivo	anti-apoptotic proteins of Bcl-2 and IAP family and activating/disturbing the mTOR/p70S6k signaling pathway/MAPKproteins(JNK and p38 MAPK)	[164]
Crocin	43 °C, 2 h	Breast cancer, MDA-MB-468/in vitro	increasing the Bax↑Bcl2↓/HSP70, HSP90↓	[167]
Perillyl alcohol	43 °C, 1 h	Mammary carcinoma, SCK/in vitro	Cell cycle arrest (TGF-b induced G1 arrest)/p53 and p21 proteins	[170]
5-aminolevulinic acid	40–46 °C	Glioma, Human grade IV GBM cell line(ACBT)/in vitro	Apoptosis	[172]
Curcumin	42 °C, 1 h	Murine Lewis lung carcinoma, MS-1/Endothelial LL/2/in vitro, in vivo	Apoptosis/angiogenesis↓	[174]
**3.3.**				
**Compound**	**HT**	**Cell Line and Observation Model**	**Molecular Mechanism**	**Ref**
Furan-fused tetracyclic compounds	44 °C, 20 min	Human lymphoma, U937/in vitro	MMP depolarization/release of cytochrome c/activating caspase-3 and 8/expression of Fas	[175]
Betulinic acid	42 °C, 2 h	Human melanoma, DB-1/in vivo	MMP depolarization	[177]
Curcumin and 5-Fluorouracil/magnetic nanoparticles encapsulated poly(D,L-lactic-co-glycolic acid)	80 °C, 60 min, 120 min	Human breast adenocarcinoma, MCF7/in vitro	destabilizing the cytoskeleton and MMP depolarization	[179]
Arsenic trioxide	43 °C, 30 min	Esophageal carcinoma, EC-1/in vitro	Cell cycle arrest G₂/M phase (and as the ratio of cells in G_0_/G_1_ and S phases decreased, cell death became more pronounced)	[182]
Curcumin and resveratrol	42 °C, 30 min	Mice colon cancer, CT26/in vitro, in vivo	inducing apoptosis/HSP70↓/recruiting CD3+ T-cells and F4/80+ macrophages	[183]
Quercetin	43 °C, 1 h	p53-negative prostatic adenocarcinoma, PC-3/Prostatic carcinoma, DU-145/in vivo, in vitro	antagonizing the expression of HSP72	[184]
Quercetin	42 °C, 1 h	Human myelogenous leukemia, K562/A, K562/in vitro	inhibition of the elevated protein expression and mRNA level of HSP70 and P-gp	[185]
Quercetin and tamoxifen	42.5 °C, 1 h	Human melanoma, M10, M14, MNT1/in vitro	reducing heat shock protein-70 expression at both protein and mRNA levels	[189]
Quercetin (HSP70 inhibitor) + LPS	NIR light at 808 nm wavelength for 5 min, 40 °C, 35 min	Human thyroid duct carcinoma, TT/in vivo, in vitro	Apoptosis (enhancement cellular uptake and pronouncement tumor targeting ability)	[190]
β-lapachone	42 °C, 1 h	Human osteosarcoma, HOS/in vitro	due to the heat-induced elevation of NQO1 activity	[191]
Enediyne	42.5 °C, 1 h	Breast cancer, MDA-231/Melanoma, U-1/in vitro	inducing DSBs, and/or a reduction in DSB repair efficiency	[193]

## 5. Conclusions and Future Direction

The schema below summarizes the combined therapy studies of HT with anti-cancer drugs or natural products (Figure 1). Several studies report that when used with anti-cancer medicines or natural products, HT significantly boosts their effects through various pathways, such as increased absorption, increased sensitivity (lower concentration), reduced side effects, and synergistic anti-cancer effects.

The principal mechanisms of HT-combined therapy involved ROS and a series of processes downstream of the ROS increase, such as DNA damage, cell cycle arrest, MMP depolarization, and decreased HSP. The uncontrolled increase in ROS concentration initiates chain reactions mediated by free radicals. ROS-targeted proteins, polysaccharides, and DNA initiate the intrinsic pathway of apoptosis. The mitochondrion is the primary source of ROS and the primary target of ROS. When the mitochondrial antioxidant system cannot maintain a stable level of free radicals, heat-induced oxidative damage affects the electron transport chain (ETC), ATP synthesis, uncoupling respiration, and the structural conformation of enzymes, lipids, and proteins [194]. As a result, an increase in ROS leads to MMP depolarization and subsequent apoptosis in cancer cells.

Furthermore, increased ROS level is linked to DNA damage. Inhibitors of the cellular heat-shock response exacerbate the impairment in DNA repair generated by HT. HT directly inhibits the DNA repair process [195] by preventing the homologous recombination repair of DNA damage [196] and DSBs [197]. When HT is combined (with anti-cancer drugs or natural products), reports indicate that cancer cell apoptosis was induced effectively through an increased ROS level or closely related mechanisms.

Cell cycle arrest is a critical component of the response mechanisms to DNA damage, and it prevents damaged cells from the division [197]. Similarly, the combined treatment of HT and anti-cancer drugs or natural products induces DNA damage and disrupts the cell cycle, resulting in cell death.

Another mechanism of the combined treatment is related to HSPs, such as HSP27, HSP70, and HSP90. HSP27 is activated by HT or anti-cancer drugs, reduces ROS levels, and prevents apoptosis via the MAPK pathway, particularly ERK. The MAPK signaling pathway plays a critical role in cell proliferation, differentiation, and survival. On the other hand, the role of ERK remains controversial because, in some systems, ERK activation can also promote apoptosis, whereas cytoprotection has been reported in others [198]. HSP70 is highly expressed in most tumor cells and acts as a survival regulator because of its anti-apoptotic properties. HSP70 inhibits both intrinsic and extrinsic apoptotic mechanisms. HSP90 is an evolutionary conserved and ubiquitously generated protein that is required for the folding, stability, activation, maturation, function, and proteolytic degradation of a variety of client proteins that are true oncoproteins involved in a variety of tumor types [199]. HSP27 and HSP90 are also important factors in cancer cell apoptosis. These HSPs protect against heat shock and reduce cell damage and ROS-mediated apoptosis. The combination with anti-cancer drugs or natural products can reduce the expression of HT-induced HSPs and allow effective HT-induced apoptosis.

Another mechanism of the combined treatments is the regulation of apoptotic factors. Related proapoptotic factors include Bax, p-53, cytochrome C, MAPKs, and AMPKα-JNK. On the other hand, Bcl-2, NF-κB, and Akt-mTOR are antiapoptotic factors, which are down-regulated in a combination treatment of HT and anti-cancer drugs or natural products. An interactive network of these pro- and anti-apoptotic factors can be regulated by combined treatment, leading to effective and synergistic cancer cell death.

The table below shows the number of studies summarized, subcategorized based on mechanisms (Table 4). By the HT therapy, there were 30 papers applying to mechanisms of ROS, 1; HSP, 2; DNA damage, 4; cell cycle arrest and differentiation, 2; regulation of the transcription factor, 6; regulation of apoptotic protein, 9; and cellular physiological changes (cytoskeleton, gene expression, collagen fiber, microvessel damage), 6. When HT was used alone, the regulation of transcription factors or apoptotic proteins was most studied. Moreover, unlike the combined therapy, six studies on the mechanism of cellular physiological changes were found. This suggests that the stress induced by heat shock may directly damage cancer cells and induce apoptosis.

There were 64 papers focusing on the co-treatment of HT with anti-cancer drugs. The detailed mechanisms were studied in the following number of papers: ROS, 6; HSP, 2; DNA damage, 5; MMP depolarization, 8; cell cycle arrest, 12; pharmacokinetics change, 7; regulation of transcription factors, 5; regulation of apoptotic proteins, 4; upregulation of p53, 3; unknown mechanisms related to apoptosis, 11; anti-angiogenesis, 1; and necrosis, 5. As indicated, ROS-related mechanisms were the most frequently investigated in the studies involving a co-treatment of HT and anti-cancer drugs. Compared to six papers on cellular physiological changes by HT, none of the combination therapy studies discussed such a mechanism. On the other hand, while none of the reports studied unknown mechanisms of apoptosis by HT alone, 11 studies were performed in a combination of HT and anti-cancer drugs. Similarly, pharmacokinetics change and anti-angiogenesis were studied only when combined therapy with HT and anti-cancer drugs were carried out. Necrosis-related studies were only performed in the case of combination with anti-cancer drugs and HT.

There were 21 papers in the case of co-treatment of HT and natural products: ROS, 7; HSP, 4; DNA damage, 1; MMP depolarization, 3; cell cycle arrest, 1; regulation of apoptotic proteins, 2; upregulation of p53, 1; and unknown mechanisms, 2. Similar to combination studies with anti-cancer drugs, the ROS-related pathway was the most frequently investigated mechanism in the combination treatment with HT and natural products. There were also four cases reporting HSP-related mechanisms, which was more than any other mechanisms besides ROS. There was no study involving the necrosis pathway, suggesting that the combination of HT with natural products can be relatively less toxic than the combination with anti-cancer drugs. Although fewer studies were reported with natural products, the increasing interest suggests the potential synergistic effects of these safe materials when combined with appropriate HT.

When anti-cancer drugs or natural products are used together with HT, mechanisms, such as increased ROS, decreased HSP, DNA damage, and MMP depolarization, were the primary mechanisms related to synergistic cell death of cancer cells. On the other hand, regulation of apoptotic transcription factors or proteins and cytoskeletal alteration were the majority mechanisms studied when HT was used alone for cancer therapy. These results indicate that the mechanisms of combined treatment are different from sole HT therapy. This difference may be because HT increases the heat-shock responses and the HSPs prevent ROS production. This review found that regulation of apoptotic factors and cytoskeletal alteration is more important on the effect of mild-HT on cancer cell apoptosis than ROS, HSP, or DNA damage. On the other hand, the combination therapy should significantly enhance the effect of HT alone because ROS is an essential pathway in cancer cell apoptosis.

Recent research also demonstrates notable development in the technology of HT. HT induction by applying magnetic nanoparticles is gaining a certain interest as an adjuvant option besides conventional chemotherapy and radiation therapy. Cancer cell-specific HT administration can be delivered by alternating magnetic fields of magnetic nanoparticles. The first attempt was made by Gilchrist et al. in 1957 demonstrated magnetic particle-induced selective heat to metastasized lymph nodes [200]. A relevant study was later carried out in the mid-90s, by injecting magnetic nanoparticles directly into cancerous regions and exciting them with magnetic fields [201]. Further effort was made on the delivery efficacy of these nanoparticles to the specific cancer site [202,203,204]. Instead of using magnetic fields, photothermally activated plasmonic nanoparticles, such as gold nanoparticles [205], GNs [206,207], GNRs [208], etc., is also another effective strategy. The detail advantages and its current challenges of nanoparticle-mediated HT can be found elsewhere [209].

Several studies investigating anti-cancer drugs or natural products focused on the increase in ROS and subsequent DNA damage and apoptosis. On the other hand, the effects are reduced gradually because HSPs and other cellular protective responses are triggered. Tumor cells can adapt to these agents, reduce the sensitivity and eventually lead to cancer recurrence. A combination of anti-cancer drugs or natural products with HT is a promising strategy to overcome this flaw. The combined therapy can control various factors that regulate the ROS levels. HT is a potent inducer of apoptosis, but the cellular responses to heat shock lower its effectiveness. Thus, a combination of HT with complementary agents, such as anti-cancer drugs or natural products (which can directly increase ROS levels and suppress protective responses), may allow the maximal delivery of apoptotic events in cancer.

## Figures and Tables

**Figure 1 antioxidants-11-00625-f001:**
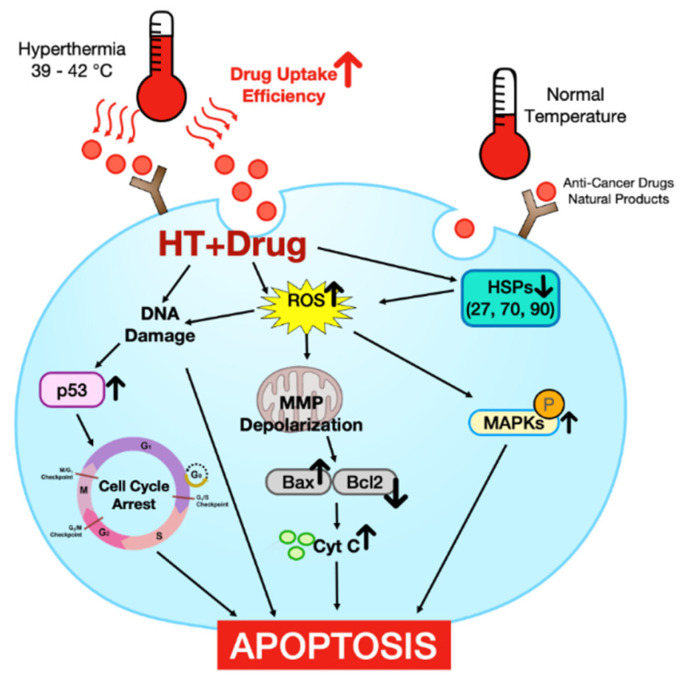
Synergic effect of HT + natural product or anticancer agent and its molecular mechanism.

**Table 4 antioxidants-11-00625-t004:** Mechanisms of HT and Its combined therapy (number of research).

Mechanism	HT	HT + Anti-Cancer Drug	HT + Natural Product
ROS	1	6	7
HSP	2	2	4
DNA damage	4	5	1
MMP depolarization		8	3
Cell cycle arrest	2	12	1
Pharmacokinetics change		7	
transcription factor	6	5	
Regulation of apoptotic protein	9	7	3
Cellular physiological changes	6		
Apoptosis (Unknown mechanism)		11	2
Anti-angiogenesis		1	
Necrosis		5

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
