# Peer review of "Hyperthermia Treatment as a Promising Anti-Cancer Strategy: Therapeutic Targets, Perspective Mechanisms and Synergistic Combinations in Experimental Approaches"

_antioxidants, 2022, doi:10.3390/antiox11040625_

Round 1

Reviewer 1 Report

The review "Hyperthermia treatment as a promising anticancer strategy: Therapeutic targets, perspective mechanisms and synergistic combinations" by Yi et al. deals with an interesting and important topic that is of interest for the scientific community.

However, there are some points, which should be improved.

Major concerns:

1) I miss an introductory section in the beginning of this review in which the authors describe how hyperthermia is or can be produced.

There is the possibility of just “increasing the temperature”, but as the authors themselves later point out, there is also “ultrasound-induced localized hyperthermia”, “photo induced hyperthermia”, “near-infrared plasmon resonance generated localized heat induced hyperthermia”, etc., etc., …

2) In addition, the complete review deals with in vitro data obtained from cell culture experiments.Or are there a few mouse models were this therapy was used? However, this

a) is not clear from the title of the publication.

b) is not formulated precisely enough in some places in the manuscript, which could give the impression that it has already been used therapeutically in humans.

c) the statement “in vivo” in tables after mentioning a cell line, is not clear for me.

What does this mean?

This has to be revised exactly!

3) The number of tables is too large. Some of the tables can be combined: e.g. Table 1 and 2. I see no reason for listing the data in separate tables.

Minor concerns:

4) Page 2, line 49: Statement: “Normal cells are not negatively affected by temperatures between 41°C and 44°C.”

This statement is in contrast to the fact that temperatures above 42°C can seriously damage proteins in the case of e.g. fever.

The statement must therefore be modified and placed in the correct context.

5) The connection to cancer is partly missing in the chapters.

In “2.2 HSPs” only hepatocytes are mentioned.

6) The introduction of abbreviations is handled differently.

In some cases, terms are abbreviated in headings. Abbreviations should always be omitted from headings and introduced for the first time in the text.

7) Page 4, line 171: HSPs was introduced as an abbreviation earlier.

8) Headline table 2: "DNA damage" does not fit in most cases with the mechanisms listed in the table.

9) Headline of “3. Anti-cancer drugs” should converted into “Combination of hyperthermia and anti-cancer drugs"

10) Page 20, line 752: Not “cH2AX” but gamma “γH2AX”!

11) Page 25, Section 4.3.1: Please specify for which type of cancer cells the therapy was used.

Reviewer 2 Report

In this review Yi and coworkers give an overview on the HT approach  for cancer treatment also indicating various combinations with cytotoxic agents as well as natural products endowed with specific properties. They eventually describe the mechanism for the improved effects of the combined tratments or their advantages. 

The topic is certainly interesting sinc HT treatmnet is important for cancer management and surly the improvement of such approach depends on the obtained experimental evidences on its effectiveness. In this view, the review well summarizes the current knowledgeand may represent an interesting starting point to continue development this therapeutic strategy. 

However, in the opinion of this referee before acceptance the ms should be improved following these points:

1) Abstract: the sentence "Several anti-cancer therapies have been identified, however further research is required to discover a perfectly safe and effective cure for cancer". I suggest to rephrase the sentence. Cancer is actually a series of different diseases with common features and a unique cure effective for all the cancers most probably will never exist.

2) Introduction: the introduction should make the reader familiar with HT approach. It should provide the bases and the main features of this approach. Here no enough information are given.

Also, in line 33-34, being cancer the 2nd cause of death it will be better to report which is the first instead of the third

line 46-47. I disagree with the fact that HT is very low cost compared to conventional therapy. Rather HT has high costs. If I'm wrong please insert relevant references.

line 64 reference needed to support the assumption.

line 123. What is E6? Please, specify

line 334-335 rephrase the sentence. Also, I disagree that carboplatin is less effective compared with cddp. Is too general. Carboplatin has an anticancer action almost superimposable to cddp in several cancers, but is better tolerated. However, despite the general consideration a strict dependency on the tumor in inherent.

line 369-370 (and subsequent section where As2O3 is cited) It should be mentioned that As2O3 (trisenox) is an approved essential drug in leukemia management.

Finally, I think authors should add a paragraph dealing with HT treatment and nanoparticles e.g. gold NPs, nanorods (NRs). Indeed this strategy may significantly improve the treatment in selected cases because of the possibility of local treatment i.e. only on the cancer tissue usig proper irradiation systems and plasmonic resonance effects. 

Overall, I suggest acceptance after the above major revision

Author Response

Dear reviewer 2,

Thank you for leaving a thoughtful comment that helped me improve my research. I'm now revising the paper in accordance with your suggestions, however I do have a question for you.

line 64 reference needed to support the assumption

I'm not sure I understand the content of your statement.
Could you perhaps explain what the assumption is and how it should be supported with references? I believe I can improve it if you notify me.

We appreciate your response.

Round 2

Reviewer 1 Report

The manuscript has been significantly improved.

All changes have been carried out to my satisfaction.

Reviewer 2 Report

Authors addressed my concerns. The ms can be now published after last check for typos and minor errors.